# DIFFERENTIAL MODEL SCALING USING DIFFERENTIAL TOPK

## ABSTRACT

Over the past few years, as large language models have ushered in an era of intelligence emergence, there has been an intensified focus on scaling networks. Currently, many network architectures are designed manually, often resulting in sub-optimal configurations. Although Neural Architecture Search (NAS) methods have been proposed to automate this process, they suffer from low search efficiency. This study introduces *Differential Model Scaling (DMS)*, increasing the efficiency for searching optimal width and depth in networks. DMS can model both width and depth in a direct and fully differentiable way, making it easy to optimize. We have evaluated our DMS across diverse tasks, ranging from vision tasks to NLP tasks and various network architectures, including CNNs and Transformers. Results consistently indicate that our DMS can find improved structures and outperforms state-of-the-art NAS methods. Specifically, for image classification on ImageNet, our DMS improves the top-1 accuracy of EfficientNet-B0 and Deit-Tiny by 1.4% and 0.6%, respectively, and outperforms the state-of-the-art zero-shot NAS method, ZiCo, by 0.7% while requiring only 0.4 GPU days for searching. For object detection on COCO, DMS improves the mAP of Yolo-v8-n by 2.0%. For language modeling, our pruned Llama-7B outperforms the prior method with lower perplexity and higher zero-shot classification accuracy.

## 1 INTRODUCTION

In recent years, large models such as GPTs (Radford et al., 2018) and ViTs (Dosovitskiy et al., 2020) have showcased outstanding performance. Notably, the emergent intelligence of GPT4 (OpenAI, 2023) has underscored the importance of scaling networks as a critical pathway toward achieving artificial general intelligence (AGI). To support this scaling process, we introduce a straightforward and potent method to determine the optimal width and depth of a network during its scaling.

Currently, the structure design of most networks still relies on human expertise. It typically demands significant resources to tune structural hyperparameters, making it challenging to pinpoint the optimal structure. Meanwhile, Neural Architecture Search (NAS) methods have been introduced to automate network structure design. We classify NAS methods into two categories based on their search strategies: stochastic search methods (Xie et al., 2022; Liu et al., 2022; Tan & Le, 2019) and gradient-based methods (Liu et al., 2018a; Wan et al., 2020; Guo et al., 2021a). The stochastic search methods involve sampling numerous sub-networks to compare performance. However, these methods are limited to low search efficiency due to the sample-evaluate cycle, leading to reduced performance and increased costs.

Unlike stochastic search methods, gradient-based methods employ gradient descent to optimize structural parameters, enhancing their efficiency and making them more adept at balancing search costs with ultimate performance. However, a significant challenge persists: how to model structural hyperparameters in **a direct and differentiable manner**. Prior methods have struggled to meet this challenge, resulting in diminished performance and increased costs. Specifically, we group prior methods into three categories based on their modeling strategies: multiple element selection, single number selection, and gradient estimate topk. Specifically, when searching for the number of channels in a convolutional layer, multiple element selection methods (Li et al.; Guo et al., 2021b) model the channel number as multiple selections of channels, as shown in Figure 1 (a.1). They introduce a much larger search space of element combinations. Single number selection methods (Wan et al.,

2020) model the channel number as a single selection from multiple numbers, as shown in Figure 1 (a.2). It ignores the order relationship among these numbers. Gradient estimate topk approaches (Guo et al., 2021a; Gao et al., 2022; Ning et al., 2020) attempt to model width and depth directly, as shown in Figure 1 (a.3). However, they are not differentiable, necessitating the development of different gradient estimation methods. As a result, these methods lack stability and are difficult to optimize.

Regrettably, all the above strategies fall short of modeling structural hyperparameters in a clear-cut and fully differentiable fashion. To address the aforementioned challenge, we introduce a fully differentiable topk operator, which can seamlessly model depths and widths in a direct and differential manner. Notably, each differential topk operator has a single learnable parameter, representing either a depth or width structural hyperparameter. It can be optimized based on guidance from both task loss and resource constraint loss. Our method stands out in terms of high optimization efficiency when contrasted with existing gradient-based approaches.

Based on our differential topk, we develop a Differential Model Scaling (DMS) algorithm to search for networks' optimal width and depth. To validate the efficacy and efficiency of our approach, we rigorously tested it across various tasks, including vision tasks and NLP tasks, and different architectures, including CNNs and Transformers.

Overall, our contributions are as follows:

- We introduce a differential topk operator, which is easy to optimize as it can model structural hyperparameters in a direct and differentiable manner.
- We develop a Differential Model Scaling (DMS) algorithm based on our differential topk to search for networks' optimal width and depth.
- We evaluate our DMS across various tasks and architectures. For example, DMS improves EfficientNet-B0 and Deit-Tiny by 1.4% and 0.6% on ImageNet, respectively, and outperforms the state-of-the-art zero-shot NAS method, ZiCo, by 0.7% while requiring only 0.4 GPU days for searching. For object detection on COCO, DMS improves the mAP of Yolo-v8-n by 2.0%. For language modeling, our pruned Llama-7B outperforms the prior method with lower perplexity and higher zero-shot classification accuracy.

## 2 RELATED WORK

The width and depth of networks are critical aspects of model architecture design. A multitude of methodologies have been proposed to automate this process, notably Neural Architecture Search (NAS) (Zoph & Le, 2016; Liu et al., 2018a) and model structure pruning (Li et al., 2020; Li et al.). NAS algorithms typically aim to design models automatically from scratch, while model structure pruning approaches focus on compressing pretrained models to enhance their efficiency. Despite their contrasting methodologies, both approaches contribute to the search for model structure.

These search methods can generally be categorized into two groups based on their search strategies: stochastic search methods (Zoph & Le, 2016; Xie et al., 2022; Liu et al., 2022) and gradient-based methods (Liu et al., 2018a; Guo et al., 2021a). In the following sections, we will introduce these methods and compare them with ours.

### 2.1 STOCHASTIC SEARCH METHODS

Stochastic search methods usually operate through a cyclical process of sampling and evaluation. At each step, they sample models with different structures and then evaluate them. This strategy is versatile as it can handle both contiguous and discrete search spaces. However, a significant downside is its low search efficiency, leading to high resource consumption and suboptimal performance. Specifically, stochastic search-based methods can be divided into three groups: multi-shot NAS, one-shot NAS, and zero-shot NAS. Multi-shot NAS (Tan & Le, 2019; Liu et al., 2022) requires the training of multiple models, which is time-consuming. For instance, EfficientNet (Tan & Le, 2019) uses over 1714 TPU days for searching. One-shot NAS (Xie et al., 2022; Cai et al., 2019) requires training a large supernet, which is also resource-intensive. For example, ScaleNet (Xie et al., 2022) uses 379 GPU days for training a supernet. Zero-shot NAS (Li et al., 2023; Lin et al., 2021) reduces

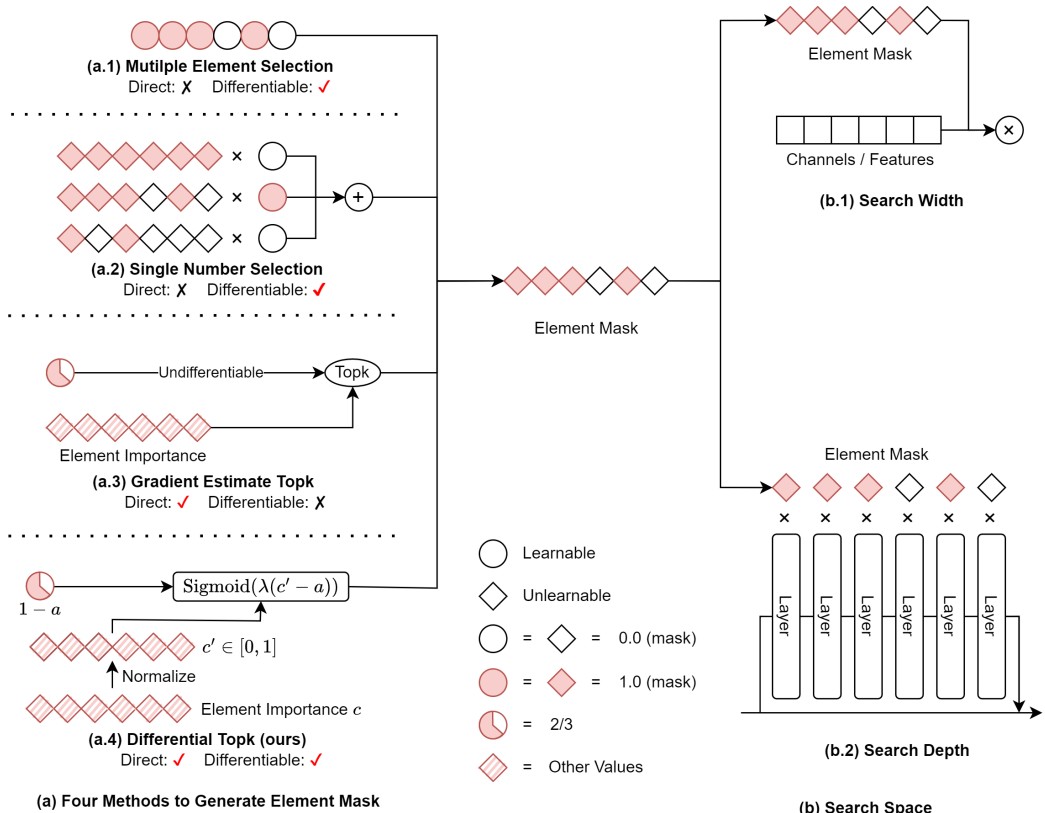

Figure 1: Different Gradient-based Modeling Strategies for Width and Depth. For all strategies, they use learnable parameters to generate an element mask to select width elements or depth elements. SubFigure **(a)** illustrates four methods to generate the element mask, while **(b)** shows how the mask is used to search width and depth. **(a.1)** Multiple Element Selections: The element count is transformed into a multiple-element selection. **(a.2)** Single Number Selections: The element count is transformed into a selection from multiple numbers. **(a.3)** Gradient Estimate Topk: The element count is directly modeled yet non-differentiable. **(a.4)** Our Differential Topk: The element count is directly modeled and is fully differentiable. "Direct" means that the learnable parameters directly model the structural hyperparameters, while "Differentiable" means that the gradient of the learnable parameters can be computed in a fully differentiable manner.

the cost by eliminating the need to train any model. However, its performance has not yet met the desired standard.

## 2.2 GRADIENT-BASED METHODS

Gradient-based structure search methods (Liu et al., 2018a; Guo et al., 2021a) employ gradient descent to explore the structure of models. Generally, these methods are more efficient than their stochastic search counterparts. The critical aspect of gradient-based methods is how to use learnable parameters to model structural hyperparameters and compute their gradients. Ideally, the learnable parameters should directly model structural hyperparameters, and their gradients should be computed in a fully differentiable manner. However, prior methods have struggled to meet these two criteria in modeling the width and depth of networks. We group them into three categories: multiple element selection, single number selection, and gradient estimate topk. The first two categories model structural hyperparameters indirectly, while the third category is not differentiable and requires gradient estimation.

Multiple element selection methods (Li et al.) model the number of elements as multiple selections from elements (e.g., channel selection), as shown in Figure 1 (a.1). They introduce a much larger

search space of element combinations. Similarly, Single number selection methods (Wan et al., 2020) model element quantity as a single choice from multiple numbers, as shown in Figure 1 (a.2). It ignores the order relationship among these numbers. These methods model structural hyperparameters in indirect and inaccurate ways, causing a gap between learnable parameters and corresponding structural hyperparameters. Naturally, They result in low performance.

Gradient estimate topk approaches (Guo et al., 2021a; Gao et al., 2022; Ning et al., 2020) attempt to model width and depth directly, as shown in Figure 1 (a.3). However, they are not differentiable, necessitating the development of different gradient estimation methods. As a result, these methods lack stability and are difficult to optimize.

To improve the optimization efficiency for structure search, we introduce a new differential topk that can model width and depth directly and is fully differentiable. Compared with previous methods, our approach reduces search costs and improves performance, demonstrating high adaptability.

## 3 METHOD

In this section, we will detail our Differential Model Scaling (DMS) in two steps. First, we introduce our differential topk, which models structural hyperparameters directly in a fully differentiable manner. Second, we explain how to use our differential topk to construct our DMS algorithm.

### 3.1 DIFFERENTIAL TOP-K

Suppose there is a structural hyperparameter denoted by $k$, representing the number of elements, such as $k$ channels in a convolutional layer or $k$ residual blocks in a network stage. $k$ has a maximal value of $N$. We use $\boldsymbol{c} \in \mathbb{R}^N$ to represent the importance of elements, where a larger value indicates a higher importance. The objective of our differential topk is to output a soft mask $\boldsymbol{m} \in [0,1]^N$ to indicate the selected elements with top $k$ importance scores.

Our topk operator uses a learnable parameter $a$ as a threshold to select elements whose importance values are larger than $a$. $a$ is able to model number of elements $k$ directly, as $k$ can be seen as a function of $a$, where $k = sum_{i=1}^{N} 1[c_i > a]$. $1[A]$ is an indicator function, which equals 1 if the A is true and 0 otherwise. We use $c_i$ to represent the importance of the $i$-th element. We denote our topk as a function $f$ as follows:

$$m_i = f(a) \approx \begin{cases} 1 & \text{if } c_i > a \\ 0 & \text{otherwise} \end{cases} \tag{1}$$

In prior methods, $f$ is usually a piecewise function, which is not smooth and differentiable, and the gradient of $a$ is computed by estimation. We argue the biggest challenge to employing a fully differentiable $f$ with respect to $a$ is that the channel importance is distributed unevenly. Specifically, uneven distribution causes the importance difference between two neighboring elements, ordered by importance value, to vary significantly. Supposed $a$ is updated by a fixed value in each iteration, when the difference is large, a lot of steps are needed for $a$ to go across these two elements. When the difference is small, $a$ can cross many elements in one step. Therefore, optimizing $a$ in a fully differentiable manner is too hard when element importance is uneven.

To address this challenge, we employ an importance normalization process to forcefully convert the unevenly distributed importance to evenly distributed values, making the topk function smooth and easy to optimize in a differentiable way. To sum up, our differential topk has two steps: importance normalization and soft mask generation.

### 3.1.1 IMPORTANCE NORMALIZATION

We normalize all element importance by mapping them to evenly distributed values from 0 to 1, based on the following:

$$c_i' = \frac{1}{N} \sum_{j=1}^{N} 1[c_i > c_j]. \qquad (2)$$

The normalized element importance is denoted by $\boldsymbol{c}'$. $1[A]$ is the same indicator function as above. Any two elements in $\boldsymbol{c}$ are supposed to be different, which is usually the case in practice. Notably, although $\boldsymbol{c}'$ is evenly distributed from 0 to 1, $\boldsymbol{c}$ can follow any distribution.

Intuitively, $c_i'$ indicates the portion of $\boldsymbol{c}$ values smaller than $c_i$. Besides, the learnable threshold $a$ also becomes meaningful, representing the pruning ratio of elements. $k$ can be computed by $k = \lfloor (1-a)N \rceil$, where $\lfloor \rceil$ is a round function. $a$ is limited to the range of $[0, 1]$, where $a = 0$ indicates no pruning and $a = 1$ indicates pruning all elements.

### 3.1.2 Soft Mask Generation

After the normalization, it's easy to generate the soft mask $\boldsymbol{m}$ using a smooth and differentiable function based on the relative size of pruning ratio $a$ and normalized element importance $\boldsymbol{c}'$.

$$m_i = f(a) = \text{Sigmoid}(\lambda(\boldsymbol{c}_i' - a)) = \frac{1}{1 + e^{-\lambda(\boldsymbol{c}_i' - a)}}. \qquad (3)$$

We add a hyperparameter $\lambda$ to control the degree of approximation from Equation 3 to a hard mask generation function. When $\lambda$ tends to infinity, Equation (3) approaches a hard mask generation function. We usually set $\lambda$ to $N$. Because when $c_i' > a + 3/N$ or $c_i' < a - 3/N$, $|(m_i - \lfloor m_i \rceil)| < 0.05$. It means that except for the six elements whose importance values are around the pruning ratio, the masks of other elements are close to 0 or 1, where the approximation error is less than 0.05. Therefore, $\lambda = N$ is sufficient to approximate a hard mask generation function for our topk.

The forward and backward graph of Equation 3 are shown in Figure 2 (a) and Figure 2 (b), respectively. It can be observed that 1) Our topk models the number of elements $k$ directly using the learnable pruning ratio $a$, and it generates a polarized soft mask $\boldsymbol{m}$ to simulate the pruned model perfectly during forward. 2) Our differential topk is fully differentiable and is able to be optimized stably. The gradient of $a$ with respect to $m_i$ is $-\lambda(1 - m_i)m_i$. Our topk intuitively detects the gradient of the mask in the fuzzy area with $0.05 < m_i < 0.95$. Note Figure 2 illustrates the gradient of $a$ with respect to $m_i$, not respect to the task loss. The gradient of $a$ with respect to the task loss is $\sum_{i=1}^{N} \frac{\partial task\_loss}{\partial m_i} \frac{\partial m_i}{\partial a}$.

### 3.1.3 Element Evaluation

As we do not limit the distribution of element importance, element importance can be quantified through various methods, such as L1-norm (Li et al., 2016), among others. In our approach, we implement Taylor importance (Molchanov et al., 2019) in a moving average manner as follows:

$$c_i^{t+1} = c_i^t \times decay + (m_i^t \times g_i)^2 \times (1 - decay). \qquad (4)$$

Here, $t$ represents the training step. $g_i$ is the gradient of $m_i$ with respect to training loss. $Decay$ refers to the decay rate. The initial value of $c_i^0$ is set to zero, and the decay rate is set to 0.99. Note that the importance of elements is not updated by gradient descent but by moving average. By leveraging Taylor importance, we can efficiently and stably estimate the importance of elements.

### 3.2 Differential Model Scaling

Relying on our differential topk, we developed Differential Model Scaling (DMS) to optimize the width and depth of networks. Our DMS follows a pipeline similar to training-based model pruning. Specifically, we begin by randomly initializing a supernet and then optimizing (or pruning) its

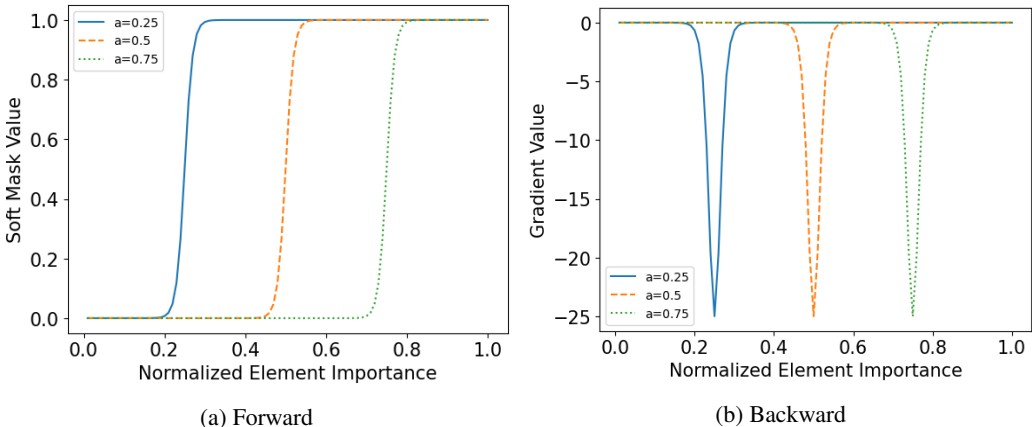

(a) Forward                            (b) Backward

Figure 2: Forwad and Backward Graph of Our Differential Top-k. We set maximal element number $N = \lambda = 100$, pruning ratio $a \in \{0.25, 0.5, 0.75\}$. The x-axis represents the normalized element importance $c'_i$. **(a)** demonstrates the forward process, where the y-axis represents the soft mask $m_i$. **(b)** illustrates the backward process, where the y-axis represents the gradient of $a$ with respect to $m_i$.

width and depth under a specific resource constraint. After this, the searched (or pruned) model is re-trained. Compared with training-based pruning, our method eliminates the need for time-consuming pretraining since we think searching from scratch is more efficient than from a pretrained model, according to our ablation study, detailed in Section 5.

As shown in Figure 1 (b), our search space encompasses both the width and depth of networks, which are the most critical structural hyperparameters for model scaling. To represent these dimensions, we use our differential topk. The width in networks typically covers the channel dimension in convolutional layers, the feature dimension in fully connected layers, and so on. Regarding depth, we focus on networks with residual connections and search the number of contiguous residual blocks in each stage. Specifically, We incorporate the soft masks of differential topk into residual connections, allowing each block to be represented as $x_{i+1} = x_i + f(x_i) \times m_i$.

To ensure that a network adheres to specific resource constraints, we incorporate an additional component into the optimization process, termed the "resource constraint loss". Consequently, the aggregate loss function is:

$$loss = loss_{task} + \lambda_{resource} \times loss_{resource}. \tag{5}$$

Here, $loss_{task}$ denotes the task loss. $loss_{resource}$ represents the additional resource constraint loss, and the term $\lambda_{resource}$ acts as its weighting factor. The resource constraint loss is further defined as:

$$loss_{resource} = \begin{cases} \log(\frac{r_c}{r_t}) & \text{if } r_c > r_t \\ 0 & \text{otherwise} \end{cases}. \tag{6}$$

In this definition, $r_c$ symbolizes the current level of resource consumption, and $r_t$ denotes the targeted level of resource consumption. $r_c$ is calculated based on the learnable parameters of differential topk operators, while $r_t$ is user-specified. As our topk is fully differentiable, the learnable structural parameters can be optimized under the guidance of both task loss and resource constraint loss.

More details about our search space and resource constraint loss are provided in Appendix A.1.

| Model | NAS Type | Top-1 | MACs | Params | Search Cost |
|---|---|---|---|---|---|
| JointPruning (Guo et al., 2021a) | Gradient | 77.3 | 0.34G | / | 0 + 8 |
| DMS-EN-350 (ours) | Gradient | **78.0** | 0.35G | 5.6M | **0 + 3.2** |
| EfficientNet-B0 (Tan & Le, 2019) | MultiShot | 77.1 | 0.39G | 5.3M | 1714 + 0 |
| DMS-EN-B0 (ours) | Gradient | **78.5** | 0.39G | 6.2M | **0 + 3.2** |
| ZiCo[‡] (Li et al., 2023) | ZeroShot | 78.1 | 0.45G | / | 0 + 0.4 |
| DMS*-EN-450 (ours) | Gradient | **78.8** | 0.45G | 6.5M | 0 + 0.4 |
| EfficientNet-B1 (Tan & Le, 2019) | MultiShot | 79.1 | 0.69G | 7.8M | 1714 + 0 |
| ScaleNet-EN-B1 (Xie et al., 2022) | OneShot | 79.2 | 0.79G | 8.3M | 379 + 3.7 |
| ModelAmplification-EN-B1 (Liu et al., 2022) | MultiShot | 79.9 | 0.68G | 8.8M | 124 + 131 |
| DMS-EN-B1 (ours) | Gradient | **80.0** | 0.68G | 8.9M | **0 + 5.8** |
| EfficientNet-B2 (Tan & Le, 2019) | MultiShot | 80.1 | 1.0G | 9.2M | 1714 + 0 |
| ScaleNet-EN-B2 (Xie et al., 2022) | OneShot | 80.8 | 1.6G | 11.8M | 379 + 7.8 |
| ModelAmplification-EN-B2 (Liu et al., 2022) | MultiShot | 80.9 | 1.0G | 9.3M | 124 + 192 |
| DMS-EN-B2 (ours) | Gradient | **81.1** | 1.1G | 9.6M | **0 + 7.0** |

Table 1: Experiments on EfficientNet. We compare our DMS with other NAS methods on Efficient-Net variants. DMS* denotes that we limit our search cost to 0.4 GPU days to compare with the zero-shot NAS method, ZiCo. The search cost associated with a model is divided into two distinct components: the public cost, like supernet training, and the private cost to search the model itself. The total search cost is represented as *public cost+private cost*. The unit of search cost is TPU days for EfficientNet and GPU days for other models. How to obtain these search costs is detailed in Appendix A.7. Besides, we provide more comparisons with NAS methods in Appendix A.2. ‡ means the model is trained with much stronger training settings than ours, such as distillation and mix-up. Note our method doesn't load pretrained weights in this table.

## 4 EXPERIMENT

We applied our method to rigorous evaluations across various tasks, including vision and NLP tasks, and architectures, including CNNs and Transformers. Notably, our method consistently outperforms both baseline models and prior NAS methods, highlighting its superior performance and adaptability.

### 4.1 EXPERIMENTS ON VISION TASKS

First, we conducted experiments on vision tasks, including image classification on ImageNet (Deng et al., 2009) and object detection on COCO (Lin et al., 2014).

#### 4.1.1 IMAGE CLASSIFICATION EXPERIMENTS ON IMAGENET

We chose a range of vision models as baselines and searched for optimal configurations in terms of their width and depth.

**EfficientNet** (Tan & Le, 2019) is a widely accepted baseline for NAS research. In our study, we revisited three variants of EfficientNet: EfficientNet-B0, B1, and B2. The performance of these searched models and their search costs are presented in Table 1.

Compared with EfficientNet, our searched models, DMS-EN-B0, B1, and B2, have improved performance by 1.4%, 0.9%, and 1.0%, respectively. Remarkably, DMS also achieves over 100 times cost savings in the search process. The searched structure of DMS-EN-B0 and EfficientNet-B0 is compared in Appendix A.8, revealing that our method applied substantial structural modifications to achieve enhanced performance.

In comparison with the multi-shot NAS method ModelAmplification (Liu et al., 2022), our method betters its performance by 0.1% and 0.2% on the B1 and B2 variants, respectively. This highlights the efficiency of our method in searching for high-performance models. ScaleNet (Xie et al., 2022), a one-shot NAS method, yields 0.7% and 0.3% lower accuracy on B1 and B2 variants despite its models being larger.

| Model | MACs | Params | mAP |
|---|---|---|---|
| Yolo-v8-n (Jocher et al., 2023) | 4.4G | 3.2M | 37.4 |
| DMS-Yolo-v8-n (ours) | 4.2G | 2.7M | **39.4** |

Table 2: Object Detection Experiments on COCO. Note our method doesn't load pretrained weights in this table.

| Model | Params | Wikitext2 ↓ | Pth ↓ | BoolQ ↑ | WinoGrande ↑ | ARC-e ↑ | ARC-c ↑ |
|---|---|---|---|---|---|---|---|
| Llama-7B (Touvron et al., 2023) | 6.74B | 12.62 | 22.14 | 76.5 | 67.01 | 72.8 | 41.38 |
| LLM-Pruner-Llama-7B (Ma et al., 2023) | 5.47B | 17.39 | 30.2 | 66.79 | 64.96 | 64.06 | 37.88 |
| DMS-Llama-7B (ours) | 5.47B | **17.13** | **27.98** | **75.23** | **65.35** | **71.46** | **39.59** |

Table 3: Experiment on Llama-7B. We pruned Llama-7B using DMS and compared it with LLM-Pruner. We evaluate the pruned model using perplexity on Wikitext2 and Pth datasets and zero-shot classification accuracy on BoolQ, WinoGrande, ARC-e, and ARC-c datasets. In our results, the symbol "↑" denotes that a larger value is better, while "↓" signifies that a smaller value is preferable. Note our method loads pretrained weight as we cannot train Llama from scratch due to resource constraints.

In comparison with both multi-shot and one-shot NAS methodologies, our approach significantly reduces the search cost. This efficiency is attributed to the flexibility of our method, allowing us to conduct model searches on a case-by-case basis without incurring a high public search cost, such as training a supernet.

ZiCo (Li et al., 2023) is an impressive zero-shot NAS method. It requires only 0.4 GPU days of search cost to achieve 78.1% top-1 accuracy with 450M MACs. In comparison, when we limit our search cost to the same 0.4 GPU days, denoted as "DMS*", our method achieves a top-1 accuracy of 78.8%, outperforming ZiCo by 0.7%.

Furthermore, aside from stochastic search methods, our DMS also excels over gradient-based methods. As exemplified by JointPruning (Guo et al., 2021a), which utilizes gradient estimation, our model, DMS-EN-350, surpasses its counterpart by a margin of 0.7% but does so with two-fifths of the search cost.

**Other Architectures**: Except for EfficientNet, we also applied our method to other architectures, encompassing ResNet (He et al., 2016), MobileNetV2 (Sandler et al., 2018), Deit (Touvron et al., 2021) and Swin (Liu et al., 2021b). The results are detailed in Appendix A.3.

### 4.1.2 OBJECT DETECTION EXPERIMENTS ON COCO

Since the complete end-to-end searching of our differential topk, DMS is a general search method that can be applied to various tasks. We also evaluated DMS for object detection on COCO. We chose Yolo-v8-n (Jocher et al., 2023) as the baseline model and searched for the optimal structure of it. Our searched version betters the original model by 2.0% in box AP, as shown in Table 2.

### 4.2 EXPERIMENTS ON LLM

Beyond vision tasks, we extended our method to evaluate its applicability on a large language model (LLM) called Llama (Touvron et al., 2023), as shown in Table 3. Due to resource constraints, we were unable to train an LLM from scratch. Instead, we adopted a "prune and finetune" strategy using the alpaca dataset (Taori et al., 2023). To mitigate overfitting to the alpaca dataset, we used the original model to distill the pruned model both during pruning and the subsequent finetuning process. In alignment with LLMPruner (Ma et al., 2023), we limited our pruning to the heads of self-attentions and the hidden dimensions of the feed-forward networks (FFN) within Llama. After pruning 20% of the parameters from Llama-7B and comparing it with LLMPruner, our method demonstrated superior performance across various benchmarks. Specifically, we observed reduced perplexity on WikiText2 (Merity et al., 2016) and Pth (Marcus et al., 1993), and higher zero-shot

| Supernet | $Iinital_{search}$ | $Iinital_{retrain}$ | $Cost_{pretrain}$ | $Cost_{search}$ | $Cost_{total}$ | Top-1 |
|----------|------------------|-------------------|-----------------|---------------|--------------|-------|
| ResNet-50 | Random | Random | 0 | 41 | 41 | 72.6 |
| ResNet-50 | Pretrain | Random | 410 | 41 | 451 | 72.5 |
| ResNet-50 | Random | Searched | 0 | 41 | 41 | 73.1 |
| ResNet-50 | Pretrain | Searched | 410 | 41 | 451 | 73.8 |
| ResNet-34 | Random | Searched | 0 | 37 | 37 | 69.4 |
| ResNet-101 | Random | Searched | 0 | 79 | 79 | 74.2 |
| ResNet-152 | Random | Searched | 0 | 116 | 116 | **74.6** |

Table 4: Ablation Study on Initialization and Structure (Supernet Size). We search for 1G models in these experiments. $Iinital_{search}$ is the initialization scheme for the search stage, $Iinital_{retrain}$ is the initialization scheme for the retrain stage, $Cost_{pretrain}$ is the cost of pretraining a supernet, $Cost_{search}$ is the cost of searching a model, $Cost_{total}$ is the total cost of pretraining and searching a model. The unit of cost is $GMACs \times epochs$.

classification accuracy on BoolQ (Clark et al., 2019), WinoGrande (Sakaguchi et al., 2021), ARC-e (Clark et al., 2018), and ARC-c (Clark et al., 2018).

In summary, our method consistently identifies high-performance model width and depth configurations compared with baseline models, outperforming prior NAS methods. It shows high performance and adaptability to various tasks and architectures. Experiment implementation details are provided in Appendix A.6. Besides, we also evaluate our method as a pure channel pruning method, which also outperforms SOTA pruning methods, detailed in Appendix A.4.

## 5 DISCUSS: STRUCTURE VS INITIALIZATION

We assessed the impact of structure and initialization on the performance of models pruned by DMS. Structure and initialization are primary determinants influencing a pruned model's performance within the research domain. However, the predominant factor among them remains unclear (Liu et al., 2018b; Frankle & Carbin, 2018). Understanding this dynamic is crucial, as it offers insights into pruning methodologies and guides future research.

Regarding initialization, we evaluated the performance of pruned models across various initialization settings, as illustrated in Table 4 (from row 1 to row 4). We employed either random initialization or pretrained initialization during the search/pruning stage and used random initialization or searched initialization during the retraining stage. For all cases, we pruned a ResNet-50 model to 1G MACs. Notably, when pretrained initialization was loaded, we used a lower learning rate to search and retrain to avoid destroying the pretrained weights. The results confirm that superior initialization significantly enhances the performance of pruned models. Both loading pretrained initialization during search and loading searched initialization when retraining can improve the performance.

Concerning structure, we derived 1G ResNet models from supernets ranging from 3.7G to 11.6 G. The corresponding results can be found in Table 4 (from row 5 to row 7). A discernible trend emerges: a larger supernet size leads to better final performance. This improvement is attributed to the broader search space available for pruned models.

In conclusion, our results indicate that both structure and initialization play a role in enhancing the performance of pruned models. However, exploring a structure with a bigger supernet is more effective and efficient than pretraining a supernet. Specifically, a model pruned from a randomly initialized ResNet-152 achieves a top-1 accuracy of 74.62%, whereas a model derived from a pretrained-initialized ResNet-50 reaches only 73.8%. Besides, the former approach involves training an 11.6G model for 10 epochs, while the latter requires training a 4.1G model for a total of 110 epochs (100 epochs for pretraining and 10 epochs for searching), nearly quadrupling the time needed. This is why DMS adheres to the pruning pipeline but omits the pretraining phase.

Besides, more ablation study about search time is detailed in Appendix A.5.

## 6 CONCLUSION

In this paper, we introduce a novel model scaling method termed *Differential Model Scaling (DMS)*. Utilizing an optimization-centric differential topk, the DMS methodically searches for the optimal width and depth of models. Compared with prior NAS methods, our DMS has three advantages. 1) DMS can identify high-performance structures, surpassing previous NAS methods. 2) DMS is cost-effective in terms of search expenses and is flexible enough to adapt to various search cost constraints. 3) DMS is universal and is compatible with a wide range of tasks and architectures.

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

## A  Appendix

### A.1  More Details about DMS

#### A.1.1  Search Space

Our search space encompasses both the width and depth of networks, which are the most critical structural hyperparameters for model scaling.

The width in networks typically covers the channel dimension in convolutional layers, the feature dimension in fully connected layers, qkv dimension and the number of heads in attention mechanisms, among others. For convolutional and fully connected layers, we use two distinct differential topk operators to model their respective input and output widths, treating each channel or feature as an individual element. For multi-head attention, we employ a single differential topk to represent the number of heads, treating each head as a separate element.

Specifically, We apply our differential topk to different layers by multiplying masks, output by differential topk operators, with inputs to layers. For convolutional layers, suppose the input is $X \in \mathbb{R}^{B \times C \times H \times W}$, and the mask is reshaped as $m \in \mathbb{R}^{1 \times C \times 1 \times 1}$, $X \times m$ works as the new input to the layer. For an attention layer, we search the head dims of qkv and the number of heads. Suppose our supernet has $H$ heads and $D$ dims in each head. We have a mask for qk head dim with $m_{qk} \in R^{1 \times 1 \times 1 \times D}$, a mask for v head dim with $m_v \in R^{1 \times 1 \times 1 \times D}$, and a mask for number of heads $m_{head} \in R^{1 \times H \times 1 \times 1}$. Suppose the sequence length is $L$, and the qkv for self-attention is $Q, K, V \in R^{B \times H \times L \times D}$. We compute the output of the self-attention by $softmax(\frac{Q'K'^T}{\sqrt{D}})V'$, where $Q' = Q \times m_{qk} \times m_{head}, K' = K \times m_{qk} \times m_{head}, V' = V \times m_v \times m_{head}$

It is crucial to highlight that there can be channel or feature dependencies within models (Liu et al., 2021a; Fang et al., 2023). Interdependent Layers are treated as one group and share the same differential topk. We implemented this using an open-source model compression toolkit MMRazor (Contributors, 2021), which is able to build element dependencies automatically.

Regarding depth, we focus on networks with residual connections. In this context, a residual block can be defined as $x_{i+1} = x_i + f(x_i)$, and contiguous residual blocks are viewed as a network stage. The depth in our approach mainly comprises the number of blocks in each stage. We use a single differential topk for a network stage, with each block functioning as a distinct element. We incorporate the soft masks of differential topk into residual connections, allowing each block to be represented as $x_{i+1} = x_i + f(x_i) \times m_i$. In the context of Transformers, an attention mechanism combined with a feed-forward network (FFN) is considered as one block sharing the same soft mask.

The depth and width structure hyperparameters are trained jointly in our approach. For example, we have a $layer$ and an input $x$; we use $m_{L_i} \in [0, 1]$ to denote the depth mask and $m_C \in [0, 1]^N$ for the width mask. The forward process is as follows: $y = m_C \times x + m_{L_i} \times layer(m_C \times x)$. After searching, we will prune depth and width according to the depth mask and width mask, respectively.

Besides, as some maximal numbers of elements are small from several to tens, like the number of blocks and attention heads, we increase the $\lambda$ in the differential topk operators from $N$ to $4N$ to approximate a hard mask generation function better.

### A.1.2   RESOURCE CONSTRAIT

Our resource constraint loss is defined as:

$$loss_{resource} = \begin{cases} \log(\frac{r_c}{r_t}) & \text{if } r_c > r_t \\ 0 & \text{otherwise} \end{cases}. \tag{7}$$

$$\tag{8}$$

In this definition, $r_c$ symbolizes the current level of resource consumption, and $r_t$ denotes the targeted level of resource consumption. If $r_c$ exceeds $r_t$, a non-zero $loss_{resource}$ is used to compress the model. The value of $r_c$ is calculated based on the learnable parameters of differential topk operators. For example, the parameter count of a fully connected layer can be computed using the formula $f_{in} \times a_{in} \times f_{out} \times a_{out}$, where $f_{in}$ and $f_{out}$ represent the number of input and output features, respectively. $a_{in}$ and $a_{out}$ are the learnable parameters of differential topk operators for that layer. The value of $r_t$ is user-specified.

To enhance stability during training, we gradually reduce $r_t$ to the final target value $r_t^{final}$ throughout the training process. For an epoch-based training procedure, $r_t$ is determined by an exponential decay function as shown below:

$$r_t = \left(\frac{r_t^{final}}{r_{supernet}}\right)^{\frac{e}{e_{max}}} \times r_{supernet}, \quad \text{where } \frac{r_t^{final}}{r_{supernet}} < 1. \tag{9}$$

Here, $e$ denotes the current epoch out of total epochs $e_{max}$. $r_{supernet}$ is a constant representing the resource demand of the supernet.

Furthermore, since resource consumption can fluctuate significantly with respect to depth, we introduce extra epochs dedicated to optimizing width while maintaining depth constant. By adopting the strategies outlined above, Differential Model Scaling ensures that models adhere to specific resource constraints.

### A.2   COMPARISON WITH MORE NAS METHODS

We additionally compare our method with more NAS methods, as shown in Table 5. Note this is a rough comparison. As some methods did not release their search costs, we simply use "High" or "Low" to represent their search costs for different NAS types. Besides, the comparison is unfair because some methods trained their models with much stronger training settings than ours. Even so, our method still outperforms these NAS methods.

Additionally, we draw Table 5 as an accuracy vs MACs plot, as shown in Figure 3. It can be observed that our method achieves comparable or better performance than other NAS methods with a low search cost level. We do not draw zero-shot NAS methods in Figure 3, as they use stronger training settings than ours, which is unfair to compare.

### A.3   IMAGE CLASSIFICATION EXPERIMENTS ON MORE ARCHITECTURES

To validate the universality of our method across various model architectures, we applied it to different architectures, as shown in Table 6.

**Classic CNNs**: We validated our method on ResNet (He et al., 2016) and MobileNetV2 (Sandler et al., 2018). Our searched ResNet surpasses ResNet-50 by 1.1%. Furthermore, when the searched ResNet is trained using an enhanced training setting (referred to as rsb-a1 (Wightman et al., 2021)), it also exceeds the corresponding model by 0.9%. Although MobileNetV2 is a lightweight model, our searched version outperforms the original model by a margin of 1.0%.

| Model | NAS Type | Top-1 | MACs | Params | Cost |
|---|---|---|---|---|---|
| MnasNet-A2 (Tan et al., 2019) | MultiShot | 75.6 | 0.34G | 4.8M | High |
| GreedyNAS-A (You et al., 2020) | OneShot | 77.1 | 0.37G | 6.5M | High |
| FBNetV2-L1 (Wan et al., 2020) | Gradient | 77.2 | 0.33G | / | Low |
| JointPruning (Guo et al., 2021a) | Gradient | 77.3 | 0.34G | / | Low |
| DMS-EN-350 (ours) | Gradient | **78.0** | 0.35G | 5.6M | Low |
| MnasNet-A3 (Tan et al., 2019) | MultiShot | 76.7 | 0.40G | 5.2M | High |
| EfficientNet-B0 (Tan & Le, 2019) | MultiShot | 77.1 | 0.39G | 5.3M | High |
| OFA (Cai et al., 2019) | OneShot | 77.7 | 0.41G | / | High |
| Zen-score‡ (Lin et al., 2021) | ZeroShot | 78.0 | 0.41G | 5.7M | Low |
| DMS-EN-B0 (ours) | Gradient | **78.5** | 0.39G | 6.2M | Low |
| BN-NAS (Chen et al., 2021a) | MultiShot | 75.7 | 0.47G | / | High |
| DONNA (Moons et al., 2021) | OneShot | 78.0 | 0.50G | / | High |
| ZiCo‡ (Li et al., 2023) | ZeroShot | 78.1 | 0.45G | / | Low |
| DMS*-EN-450 (ours) | Gradient | **78.8** | 0.45G | 6.5M | Low |
| EfficientNet-B1 (Tan & Le, 2019) | MultiShot | 79.1 | 0.69G | 7.8M | High |
| Zen-score‡ (Lin et al., 2021) | ZeroShot | 79.1 | 0.60G | 7.1M | Low |
| ScaleNet-EN-B1 (Xie et al., 2022) | OneShot | 79.2 | 0.79G | 8.3M | High |
| ModelAmplification-EN-B1 (Liu et al., 2022) | MultiShot | 79.9 | 0.68G | 8.8M | High |
| DMS-EN-B1 (ours) | Gradient | **80.0** | 0.68G | 8.9M | Low |
| EfficientNet-B2 (Tan & Le, 2019) | MultiShot | 80.1 | 1.0G | 9.2M | High |
| ZiCo‡ (Li et al., 2023) | ZeroShot | 80.5 | 1.0G | / | Low |
| ScaleNet-EN-B2 (Xie et al., 2022) | OneShot | 80.8 | 1.6G | 11.8M | High |
| Zen-score‡ (Lin et al., 2021) | ZeroShot | 80.8 | 0.9G | 19.4M | Low |
| ModelAmplification-EN-B2 (Liu et al., 2022) | MultiShot | 80.9 | 1.0G | 9.3M | High |
| BigNAS-XL (Liu et al., 2022) | OneShot | 80.9 | 1.0G | 9.5M | High |
| DMS-EN-B2 (ours) | Gradient | **81.1** | 1.1G | 9.6M | Low |

Table 5: Rough Comparison with more NAS methods. As some methods did not report their search cost, we simply use "High" and "Low" to represent the search cost of NAS methods, while "High" for multi-shot NAS methods and one-shot NAS methods, "Low" for gradient-based NAS methods and zero-shot NAS methods. ‡ means the model is trained with much stronger training settings than ours, such as distillation and mix-up. Note our method does not load pretrained weights in this table.

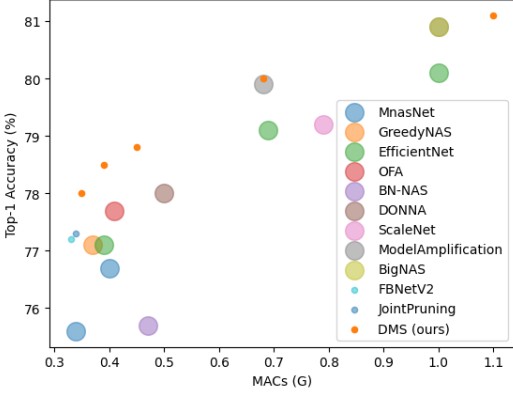

Figure 3: Accuracy vs MACs Plot. We draw this plot based on Table 5. We use two dot sizes to represent the "High" and "Low" search cost levels. Our method achieves comparable or even better accuracy with a low search cost level compared with other methods with a high search cost level.

| Model | Top-1 | MACs | Params |
|---|---|---|---|
| ResNet-50 (He et al., 2016) | 76.5 | 4.1G | 25.6M |
| DMS-ResNet | **77.6** | 4.0G | 28.4M |
| ResNet-50-rsb-a1 (He et al., 2016) | 80.1 | 4.1G | 25.6M |
| DMS-ResNet-rsb-a1 | **81.0** | 4.0G | 28.4M |
| MobileNetV2 (Sandler et al., 2018) | 72.0 | 0.3G | 3.4M |
| DMS-MobileNetV2 | **73.0** | 0.3G | 5.3M |
| Deit-T (Touvron et al., 2021) | 74.5 | 1.3G | 5.7M |
| DMS-Deit-T | **75.1** | 1.3G | 6.2M |
| Swin-T (Liu et al., 2021b) | 81.3 | 4.5G | 29M |
| DMS-Swin-T | **81.6** | 4.6G | 44.6M |

Table 6: Experiments on ImageNet with Various Architectures. We searched the models' width and depth and compared them with the original models. Note our method doesn't load pretrained weights in this table.

| Method | MACs | Top-1 |
|---|---|---|
| ResNet-50 | 4.1G | 76.5 |
| LFPC (He et al., 2020) | 1.6G | 74.46 |
| GReg2 (Wang et al., 2020) | 1.6G | 74.93 |
| CC (Li et al., 2021) | 1.5G | 74.54 |
| TPP (Wang & Fu, 2022) | 1.6G | 75.12 |
| DMS(ours) | 1.6G | **75.53** |

Table 7: Comparison with SOTA Pruning Methods. We prune ResNet-50 with pretrained initialization and compare our method with SOTA pruning methods. Our method loads pretrained weights as other pruning methods.

**Transformers**: We additionally applied our method to Transformers, encompassing Deit (Touvron et al., 2021) and Swin (Liu et al., 2021b). Deit is a one-stage Transformer, while Swin is a four-stage Transformer. Although they have different architectures, our searched models outperform the original models by 0.6% and 0.3%, respectively.

## A.4    COMPARISON WITH SOTA PRUNING METHODS

We compared our method with SOTA pruning methods in Table 7. Following structure pruning methods, we only pruned channels in ResNet-50 with the pretrained initialization and then finetuned the pruned model with the same training setting as TPP (Wang & Fu, 2022).

Compared with SOTA pruning methods, our DMS achieves the best performance even though we do not employ complicated channel importance evaluation methods like others. This is because of the strong ability of our DMS to search for the optimal structure of models.

## A.5    MORE ABLATION STUDY

In this section, we conduct more ablation studies to help readers better understand our method. We use ResNet-50 as our supernet and search models with 1G MACs by default.

### A.5.1    ABLATION STUDY OF SEARCH TIME

We assessed the relationship between search time and final performance, as detailed in Table 8. The results indicate that extended search durations typically yield superior structures and better performance. Besides, even though the search time is only 3 epochs, the performance of the searched model is still better than ResNet-18.

| Search Time | MACs | Top-1 |
|---|---|---|
| ResNet-18 | 1.8G | 69.9 |
| 3 epochs | 1G | 71.6 |
| 5 epochs | 1G | 72.9 |
| 10 epochs | 1G | **73.1** |
| 20 epochs | 1G | 72.8 |

Table 8: Ablation Study on Search Time. We compare the performance of models with different search time. We do not load pretrained weights in this table.

| Method | Search Space | Top-1 | Search Cost |
|---|---|---|---|
| Autoformer-T (Chen et al., 2021b) | Autoformer | 74.7 | > 25 GPU days |
| DMS (ours) | Autoformer | **75.2** | 2 GPU days |

Table 9: Ablation Study on Search Space. We compare the performance with Autoformer with the exact same search space. We do not load pretrained weights in this table.

### A.5.2    ABLATION STUDY OF SEARCH SPACE

Due to the high search efficiency of our method, our method can deal with a much fine-grained search space, while stochastic search methods usually use a course-grained search space due to their low search efficiency. Specifically, for a structure hyperparameter $x$, we directly search it in the range of $[1, x_{max}]$ with step 1, while stochastic search methods usually search it in the range of $[x_{min}, x_{max}]$ with step larger than 1, such as 32 and 64. Therefore, our search space is much larger and easier to design than stochastic search methods.

To compare stochastic search methods fairly, we search a transformer model with the exact same search space as Autoformer (Chen et al., 2021b), a robust stochastic search method. The results are shown in Table 9. Our method achieves better performance than Autoformer with the exact same search space. Besides, our method only takes 2 GPU days to search, while Autoformer takes more than 25 GPU days.

### A.5.3    ABLATION STUDY OF ELEMENT IMPORTANCE METRIC

Here, we provide an ablation study of the element importance metric. We design a basic element importance metric, index metric. The index metric statically assigns importance values according to the index of elements. Specifically, we set a smaller importance value for an element with a smaller index, while the width elements on the left have a smaller index than that on the right, and the depth elements closer to the input of the model have a larger index than those closer to the output of the model. As we use Taylor importance (Molchanov et al., 2019) as our default metric, we also provide a comparison of Taylor importance with and without a moving average.

The results are shown in Table 10. Taylor importance is better than the index metric as it's able to assign importance value dynamically. Besides, the moving average can improve performance by smoothing the update of importance values.

| Element Importance | Top-1 |
|---|---|
| Index metric | 72.3 |
| Taylor importance, without moving average | 72.5 |
| Taylor importance, with moving average | **73.1** |

Table 10: Ablation Study on Element Importance Metric. Index metric assigns importance values statically according to the index of elements. We do not load pretrained weights in this table.

| $\lambda_{resource}$ \ $lr_{structure}$ | 5e-2 | 5e-3 | 5e-4 | 5e-5 |
|---|---|---|---|---|
| 0.1 | / | / | / | / |
| 1 | 73.0 | **73.1** | 72.9 | / |
| 10 | 72.5 | 72.2 | 72.6 | 70.9 |

Table 11: Ablation Study on Unfixed Hyperparameters. "/" denotes the model is not able to reach our resource target. We do not load pretrained weights in this table.

### A.5.4 ABLATION STUDY OF HYPERPARAMETERS OF OUR METHOD

We divide the hyperparameters of our method into two categories: fixed hyperparameters and unfixed hyperparameters. Fixed hyperparameters are hyperparameters that are fixed for all models, while unfixed hyperparameters are hyperparameters that are turned for different models.

The fixed hyperparameters include the decay rate for Taylor importance and the temperature $\lambda$ for our differential topk operator.

Taylor importance (Molchanov et al., 2019) is a well-known method to measure the importance of elements, and the decay of moving average is also widely used in the literature. Therefore, we directly use the decay rate of 0.99 regarding prior works.

Temperature $\lambda$ of our diffenretial topk. The temperature is used to polarize (Zhuang et al., 2020) the mask of elements. Directly selecting a value that can polarize the mask of elements is enough. Thanks to our importance normalization, the temperature can be directly computed by closed-form, detailed in Section 3.1.2. The temperature $\lambda$ is set to $N$ for width elements and $4N$ for depth elements and the number of heads in attention mechanisms. $N$ is the number of elements in the corresponding dimension. They work well for all models.

Therefore, we do not conduct an ablation study on these fixed hyperparameters.

The unfixed hyperparameters include the weight of resource constraint loss $\lambda_{resource}$ and the learning rate for structure parameters $lr_{structure}$. They are used to control the update of the structure parameters. The update value of a structure parameter is computed by $lr_{strucutre} \times (g_{task} + \lambda_{resource} \times g_{resource})$, where $g_{task}$ and $g_{resource}$ is the gradient of structure parameters with respect to the task loss and resource constraint loss, Table 11 shows the ablation study results.

Obviously, 1) Smaller $\lambda_{resource}$ is better, as far as the model can reach the target resource constraint. Smaller $\lambda_{resource}$ means that the task loss takes more control of the update of the structure parameter. 2) When $\lambda_{resource}$ is small, the model is not sensitive to the change of $lr_{structure}$. When $\lambda_{resource}$ is large, a relatively large $lr_{structure}$ is better. This is because reaching the target resource constraint can reduce the influence of the resource constraint loss, as resource constraint loss is zero when the model reaches the target resource constraint.

Therefore, the setting of $\lambda_{resource}$ and $lr_{structure}$ is not difficult. We first fix $lr_{structure}$ and turn $\lambda_{resource}$ to a small value and ensure the model can reach the target resource constraint. Then, we turn $lr_{structure}$ to a relatively large value, which makes the model reach the target resource constraint in the first hundreds of iterations. Only observing the resource decrease in the first epoch is enough to set these two hyperparameters.

Compared with other NAS methods, our method uses fewer hyperparameters. For example, ModelAmplification (Liu et al., 2022) must turn at least five hyperparameters for different tasks and models.

### A.6 DETAIL OF EXPERIMENT SETTING

In general, given a baseline model and a training setting, we enlarge the baseline model as our supernet and decrease the number of epochs of the training setting as our pruning setting. We list details of our experiment setting as shown below.

**EfficientNet**: For all DMS-ES variants, we pruned the supernets over a span of 30 epochs. For those DMS-ES variants with MACs fewer than 0.5G, the pruning was conducted from EfficientNet-B4, using an input size of 224. Meanwhile, for DMS-EN-B1 and B2, the pruning was initiated from EfficientNet-B7. The input sizes for DMS-EN-B1 and B2 were 256 and 288, respectively. Subsequently, the DMS-EN variants were retrained using the corresponding training scripts of EfficientNet available in the Timm library (Wightman, 2019).

**ResNet**: We pruned the ResNet over ten epochs, starting from the ResNet-152 model. After pruning, the ResNet was retrained utilizing the MMPretrain (Contributors, 2023) training settings. This encompasses the foundational setting with a step learning scheduler and the rsb-a1 configuration.

**MobileNetV2**: To search for the ideal structure for MobileNet, we commenced by enlarging MobileNetV2 before pruning. Specifically, all channel numbers were expanded by 1.5 times, and the number of blocks in each stage was doubled. The pruning process for MobileNetV2 spans 30 epochs. Subsequent to this, the architecture was retrained employing the MMPretrain training settings.

**Deit**: We enhanced the depth of the Deit-small model, moving from 12 to 16, to serve as the supernet. The pruning for Deit was conducted with 30 epochs, including 20 epochs as a warmup phase and the model was fixed. After pruning, we retrained the model using MMPretrain combined with the Swin training setting.

**Swin Transformer**: To form our supernet, we augmented the Swin-Tiny by doubling the blocks in each stage and boosting the embedding dimension from 96 to 128. The pruning is executed over 30 epochs, including 20 epochs as a warmup phase. Once pruned, the model was subsequently retrained using MMPretrain.

**Yolo-v8** We used Yolo-v8 with deepen factor of 0.5 and widen factor of 0.5 as our supernet, while the original Yolo-v8-n has deepen factor of 0.33 and widen factor of 0.26. We used the training setting of Yolo-v8-n to train the supernet and pruned it over 30 epochs. The experiment of Yolo-v8 was conducted based on MMYolo (Contributors, 2022).

## A.7 DETAIL OF SEARCH COST ESTIMATION

In this section, we delve into the specifics of how we estimate the search costs for other NAS methods as outlined in Table 1. The search cost of a searched model is divided into two parts: the public part and the private part. The public part is conducted for all sub-models, while the private part pertains to a specific sub-model.

**EfficientNet**: EfficientNet searches for common scaling strategies across all variants, thus incurring no private search cost. The public search cost estimate for EfficientNet is sourced directly from the ScaleNet paper (Xie et al., 2022).

**ScaleNet** (Xie et al., 2022): The ScaleNet paper explicitly presented their search cost, which includes a public cost of 379 GPU days and a private cost of 106 GPU days for several sub-models, totaling 21G MACs. We compute the private search cost for a sub-model based on the ratio of its MACs to the overall 21G MACs.

**ModelAmplification** (Liu et al., 2022): As a multi-shot NAS method, ModelAmplification requires training multiple models. For all sub-models, it utilizes a public proxy dataset and a proxy training script. Approximately 2007 epochs are expended to examine the proxy dataset, and an additional 2963 epochs are used for the proxy training script, leading to a total of 4970 epochs. During the model search phase, for a variant with 390M MACs, ModelAmplification trains about 390 models per iteration. Assuming a ten-fold iteration search per model, this results in roughly 3000 epochs. By benchmarking the training time of EfficientNet-B0 on A100, we determine that 100 epochs require about 2.5 GPU days. As a result, the public search cost for ModelAmplification is at least 144 GPU days, while the private cost for the 390M MACs variant is 75 GPU days. We linearly scale the search costs of different variants based on their MACs.

**JointPruning** (Guo et al., 2021a): As a gradient-based pruning method, JointPruning presumably employs a supernet and training script analogous to ours. We deduce its search cost based on the number of pruning epochs. JointPruning paper indicates that a quarter of the total training epochs

is earmarked for model searching. In contrast, we utilize at most a tenth of the total epochs for this purpose. Hence, the search cost for JointPruning is 2.5 times that of ours.

## A.8 VISUALIZATION OF SEARCHED MODEL STRUCTURE

In Figure 4, a visualization is provided to delineate the structural intricacies of our searched DMS-EN-B0 in comparison to EfficientNet-B0. A distinct observation that stands out is the depth of our DMS-EN-B0. It possesses 8 more inverted residual blocks than its EfficientNet counterpart. Furthermore, when we delve deeper into the channel distribution across different stages, it becomes evident that our DMS-EN-B0 has undergone significant structural modifications, veering away from the traditional blueprint of EfficientNet-B0. Such distinct differences underscore the fine-grained adaptability of our method, emphasizing its capability to recalibrate and refine models in a way that they are acutely tailored to the task.

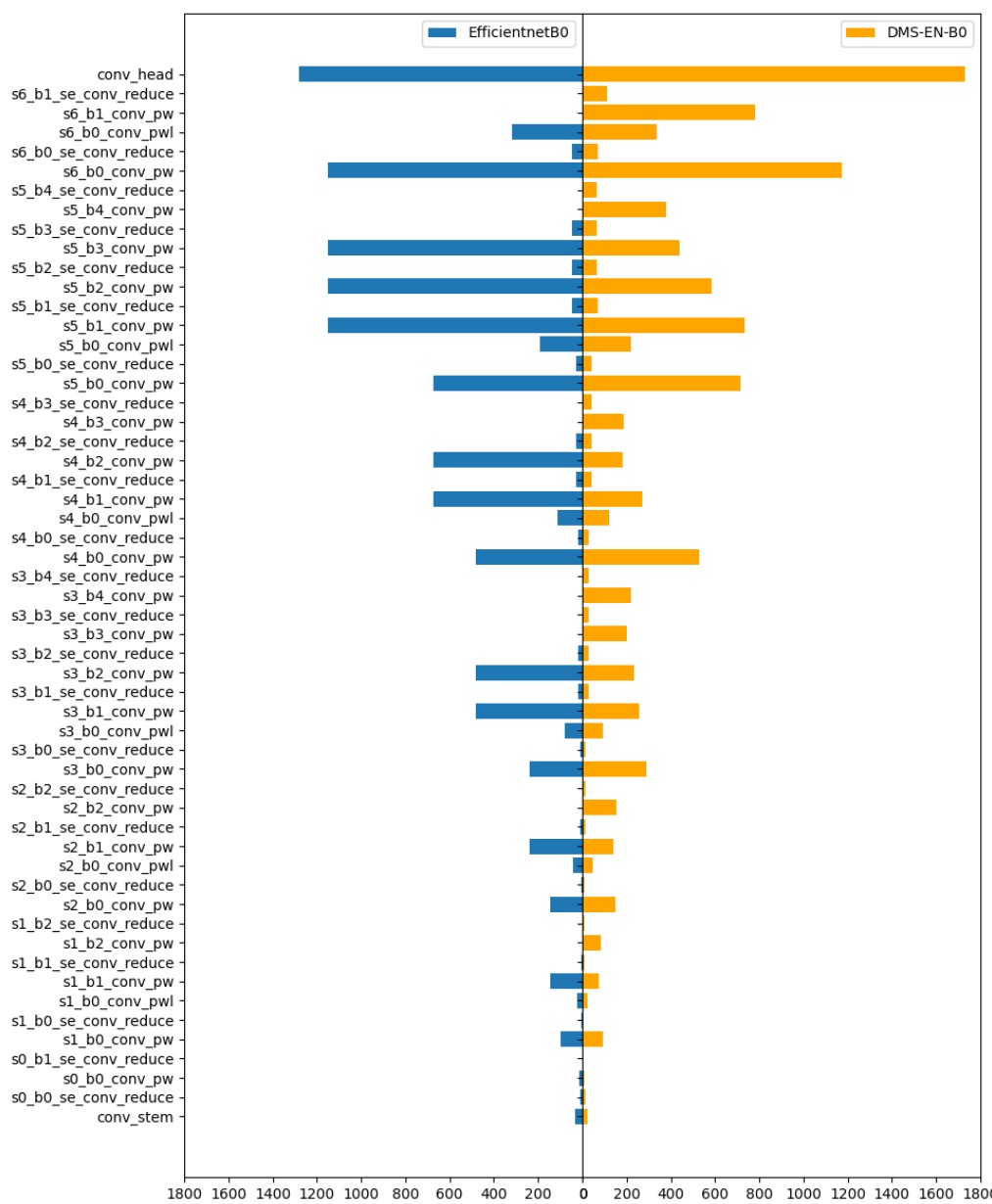

Figure 4: Visualization of Our Searched Structure. The x-axis represents the layers' width (channels/features), while the y-axis represents the layers. As DMS-EN-B0 has more layers than EfficientNet-B0, the width of extra layers for EfficientNet-B0 are seen as 0.

