# OpenReview forum: "Differential Model Scaling using Differential Topk"
_ICLR.cc/2024/Conference — Submitted to ICLR 2024_

### Official Review · Reviewer_L4n8 · 2023-10-29

**Soundness:** 3 good
**Presentation:** 4 excellent
**Contribution:** 2 fair
**Rating:** 5
**Confidence:** 5

**Summary:**

This paper proposes DMS which can find improved structures and outperforms state-of-the-art NAS methods. It demonstrated improved performance on image classification, object detection and LLM.

**Strengths:**

The proposed method is sound and straightforward.

**Weaknesses:**

- The image classification baselines are too weak. baselines should have 90%+ top-1 accuracy.
- Shown in Table 6, the performance gain is marginal.

**Questions:**

- accuracy vs MACs Plot with Table 6. The performance gain seems marginal from the table.
- explain how this loss_resource is designed.

---

> ### Author Response · Authors · 2023-11-16
> **Response to Reviewer L4n8**
>
> Dear reviewer,
>
> Thank you for your comments. We have addressed all of them below.
>
> > W1 "The image classification baselines are too weak. baselines should have 90%+ top-1 accuracy."
>
> Thanks for your suggestion. We agree with you that a strong baseline is very important.
>
> However, in this paper, our main target is to compare the search efficiency of our method and other search methods. Hence, we select our baselines from the related work for a fair comparison. All of the most related work, including ScaleNet, ModelAmplification, and so on, use EfficientNet as their baselines.
>
> Since we think our baselines are suitable for our target rather than too weak. We will consider using stronger baselines in our future work if we need to build a SOTA image classification model.
>
> > W2 "Shown in Table 6, the performance gain is marginal.",
> > Q1 "accuracy vs MACs Plot with Table 6. The performance gain seems marginal from the table."
>
> Sorry for the confusion. There are two reasons why our performance gain seems 'marginal'.
>
> 1. Compared with the Multi-Shot and One-shot methods, our methods have a much lower search cost, up to several tens, even hundreds of times lower. Therefore, our performance gain is not marginal because we may use much lower search costs to achieve a higher performance.
> 2. Compared with zero-shot NAS methods, they all use much stronger training recipes than ours, such as distillation and mixup. Therefore, the real improvement gain is more significant than the table shows.
>
> Here, we provide a new table to make the comparison clearer (We only list some methods that may be confusing; please refer to Table 5 in our revised paper for full comparison.)
>
> | Method                   | NAS Type  | Top1     | MACs  | Params | Cost |
> | ------------------------ | --------- | -------- | ----- | ------ | ---- |
> | Zen-score (+)            | ZeroShot  | 78.0     | 0.41G | 5.7M   | Low  |
> | **DMS-EN-B0 (ours)**     | Gradient  | **78.5** | 0.39G | 6.2M   | Low  |
> | Zen-score （+）          | ZeroShot  | 79.1     | 0.60G | 7.1M   | low  |
> | ModelAmplification-EN-B1 | MultiShot | 79.9     | 0.68G | 8.8M   | High |
> | **DMS-EN-B1 (ours)**     | Gradient  | **80.0** | 0.68G | 8.9M   | Low  |
> | ScaleNet-EN-B2           | OneShot   | 80.8     | 1.6G  | 11.8M  | High |
> | Zen-score (+)            | ZeroShot  | 80.8     | 0.9G  | 19.4M  | Low  |
> | ModelAmplification-EN-B2 | MultiShot | 80.9     | 1.0G  | 9.3M   | High |
> | BigNAS-XL                | OneShot   | 80.9     | 1.0G  | 9.5M   | High |
> | **DMS-EN-B2 (ours)**     | Gradient  | **81.1** | 1.1G  | 9.6M   | Low  |
>
> (+) means the model is trained using much stronger training tricks, such as distillation and mixup. As some methods do not report their search costs, we use "High" and "Low" to provide a rough comparison.
>
> Thanks for your comments. We updated the table (Note the index is 5 in our revised paper, while it's 6 in the original paper).
> We also added an accuracy vs MACs plot to the paper, notated as Figure 3 in our revised paper.
>
> > Q3 "explain how this loss_resource is designed.
>
> Resource constraint loss is widely used in the field of NAS and pruning [1,2]. We just follow the common practice in the field.
>
> Take constraining the number of parameters of a fully connected layer as an example. Suppose the layer has $f_{in} \in R$ input features and $f_{out} \in R$ output features. The structure parameters for its input features and output features are $a_{in} \in [0,1]$ and $a_{out} \in 0,1$, respectively. The resource constraint loss for the layer is defined as $f_{in} \times a_{in} \times f_{out} \times a_{out}$. The resource constraint loss of the total model is the sum of all parameter losses of all layers.
>
> We also detailed the resource constraint loss in Appendix A.1.2. Please refer to it for more details.
>
> [1] Li, Yanyu, et al. "Pruning-as-Search: Efficient Neural Architecture Search via Channel Pruning and Structural Reparameterization."
> [2] Gao, Shangqian, et al. "Disentangled differentiable network pruning." European Conference on Computer Vision. Cham: Springer Nature Switzerland, 2022.
>
>
> Thanks for your comments again. If you have any further comments, please let us know.
> Besides, we want to emphasize the value of our method. We build a general and flexible NAS method. It can improve various tasks stably with a reasonable search cost, while prior methods are usually designed for a specific task and use much more resources. Our method pushes the boundary of NAS and makes it more practical for real-world applications.
> Therefore, we think our method is valuable and worth publishing. I hope you can raise our paper's score if our responses are convincing. Thank you very much.

---

> ### Author Response · Authors · 2023-11-21
> **Sincerely expecting feedback from reviewer L4n8.**
>
> Dear reviewer:
>
> Thanks for your constructive comments. We have posted our responses to your comments. We expect your feedback about whether our responses address your concerns, or if you have any further questions. We are glad to answer them and improve our paper.
>
> Best,
>
> Authors

---

### Official Review · Reviewer_9aAa · 2023-10-30

**Soundness:** 2 fair
**Presentation:** 3 good
**Contribution:** 3 good
**Rating:** 6
**Confidence:** 3

**Summary:**

The paper introduces a new method called Differential Model Scaling (DMS) for optimizing network architectures, resulting in improved performance across diverse tasks. The study addresses the inefficiency of existing Neural Architecture Search (NAS) methods and proposes DMS as a more efficient alternative for searching optimal width and depth in networks.

**Strengths:**

- The paper introduces a novel, fully differentiable model scaling method, addressing a fundamental challenge in neural network architectures.
-  The developed search algorithm efficiently identifies optimal network structures, potentially reducing computational costs in architecture search.
-  The paper is well-written, with a clear and accessible style that enhances understanding, making it broadly accessible to the scientific community.

**Weaknesses:**

- The paper does not provide the code or the implementation details of the proposed method, which makes it difficult to reproduce and verify the results.
- The paper does not explain how the layerwise mask affects the channel-wise mask in the differential topk. It is unclear how the two masks interact and whether they can be jointly optimized in an efficient way.
- The paper lacks a proper control experiment to isolate the effect of the differential topk from other factors, such as the network architecture, the learning rate, and the data augmentation. It is possible that some of the improvements are due to these factors rather than the proposed method.
- The paper introduces too many hyperparameters for the differential topk, such as the temperature, the sparsity ratio, and the regularization coefficient. The paper does not provide a systematic analysis of how these hyperparameters affect the performance and the stability of the method. It is also unclear how to choose these hyperparameters for different tasks and architectures.
- The paper's ablation study is not comprehensive enough to demonstrate the advantages of the proposed method. The paper only compares the differential topk with a uniform scaling baseline, but does not compare it with other model scaling methods, such as compound scaling or progressive scaling. The paper also does not show how the differential topk performs on different network layers, such as convolutional layers or attention layers.

**Questions:**

- In Section 3.1.3, you use a moving average to update the layerwise mask. What is the motivation and benefit of this technique? How does it affect the convergence and stability of the optimization?
- In Section 3.1.3, you adopt an L1-norm regularization term for the channel-wise mask. Why did you choose this norm over other alternatives, such as L2-norm or entropy? How does the choice of norm influence the sparsity and diversity of the channel-wise mask?

---

> ### Author Response · Authors · 2023-11-16
> **Response to Reviewer 9aAa (Part 1/3)**
>
> Dear reviewer:
>
> Thanks for your comments. Below are our responses to address your concerns.
>
> > W1 "The paper does not provide the code or the implementation details of the proposed method, which makes it difficult to reproduce and verify the results."
>
> Considering the double-blind review policy, we can not open-source our code before the paper is accepted. However, we will open-source our code after the paper is accepted.
>
> > W2 "The paper does not explain how the layerwise mask affects the channel-wise mask in the differential topk. It is unclear how the two masks interact and whether they can be jointly optimized in an efficient way."
>
> The layerwise mask and the channel-wise mask are independent.
> For example, we have a $layer$ and an input $x$. We use $m_{L_i} \in [0,1]$ to denote the layerwise mask and $m_{C}\in [0,1]^{N}$ for the channel-wise mask.
> The forward process is as follows: $y= m_C \times x+m_{L_i} \times layer(m_C \times x)$.
> After searching, we will prune layers and channels according to the layerwise and channel-wise masks, respectively.
>
> We updated Appendix A.1.1 to make it more clear.
>
> > W3 "The paper lacks a proper control experiment to isolate the effect of the differential topk from other factors, such as the network architecture, the learning rate, and the data augmentation. It is possible that some of the improvements are due to these factors rather than the proposed method."
>
> We think we have controlled the factors you mentioned.
> For architecture, we evaluate our method always with the same architecture as our baselines.
> For learning rate and data augmentation, without extra explanation, we use the same training setting for our method and the baseline methods.
> We detail the training setting in Appendix a.6. If you think where we have not controlled the factors, please let us know.

---

> > ### Comment · Reviewer_9aAa · 2023-11-20
> > **About W2**
> >
> > I am still confused about W2. From your equation, $m_{L_i}\times layer(m_{C}\times x)$, I think the update of $m_{L_i}$ is related to that of $m_{C}$.

---

> > > ### Author Response · Authors · 2023-11-20
> > > **Responds to "About W2"**
> > >
> > > There are two processes related to the mask.
> > > 1. During training, the masks work as polarized soft gates. The pruned part will be reset to zero but not really pruned. Therefore, the masks are updated independently.
> > > 2. After training, we need to export the searched model. We first prune channels and then prune depth(layers). All masks are converted to hard masks by $m^{\prime} = \mathbb{I}(m > 0.5)$. The steps are as follows:
> > >    1. First, we prune channels. We prune $layer$ according to $m_C$, and get $layer^{\prime}$. We also prune $x$ according to $m_C$, and get $x^{\prime}$. The model becomes $x^{\prime}+ m_{L_i}$$\times layer^{\prime}(x^{\prime})$.
> > >    2. Then, we prune the depth. If $m_{L_i} = 0$, the layer is removed and directly returns $x^{\prime}$. Otherwise, we return $x^{\prime}+layer^{\prime}(x^{\prime})$.
> > >
> > > As above, we do not directly merge $m_C$ and $m_{L_i}$, so they are independent.
> > >
> > > We guess you are confused about whether $m_{C}$ equals 0 when $m_{L_i}=0$.
> > > In logic, $m_{C}$ is absolutely equal to 0 when when $m_{L_i}=0$.
> > > In practice, there is no need to set $m_{C}$ to zero when $m_{L_i}=0$, as we prune channels first and then prune the layer. Whatever the value of $m_{C}$ is, if $m_{L_i}=0$, the layer, including all channels(filters), will be removed.
> > >
> > > We hope our explanation is clear. If you still have any questions, please feel free to contact us.

---

> ### Author Response · Authors · 2023-11-16
> **Response to Reviewer 9aAa (Part 2/3)**
>
> > W4 "The paper introduces too many hyperparameters for the differential topk, such as the temperature, the sparsity ratio, and the regularization coefficient. The paper does not provide a systematic analysis of how these hyperparameters affect the performance and the stability of the method. It is also unclear how to choose these hyperparameters for different tasks and architectures."
>
> Sorry for the confusion. We think our hyperparameters are not too many, as there are only two hyperparameters that need to be turned for different tasks and architectures, and the other hyperparameters are fixed.
>
> We detail our hyperparameters as follows:
>
> We first introduce the fixed hyperparameters.
>
> 1. Decay for Taylor importance. Taylor importance[1,2] is a well-known method to measure the importance of elements, and the decay of moving average is also widely used in the literature. Therefore, we directly use the value from prior work.
> 2. Temperature of our diffenretial topk. The temperature is used to polarize [3] the mask of elements. This is also a widely used approach for pruning. Directly selecting a value that can polarize the mask of elements is enough. Thanks to our importance normalization, the temperature can be directly computed by closed-form, detailed in section 3.1.2. Therefore, it's also fixed.
> 3. Supernet size and sparsity ratio. In our opinion, the supernet size and sparsity ratio are not hyperparameters. They are set according to the user's demand and their resource amount.
>
> Only two hyperparameters need to be turned for different tasks and architectures.
> 1. Resource constraint loss weight, denoted by $\lambda_{resource}$
> 2. Learning rate for structure parameters denoted by $lr_{structure}$
>
> They are used to control the update of the structure parameters. The update value of a structure parameter is computed by $lr_{strucutre}\times (g_{task} + \lambda_{resource}\times g_{resource})$, where $g_{task}$ and $g_{resource}$ is the gradient of structure parameters with respect to the task loss and resource constraint loss,
> We provide an ablation study as follows:
>
> | - | 5e-2 | 5e-3 | 5e-4 | 5e-5 |
> |--|------|------|------|------|
> | 0.1 | / | / | / | / |
> | 1 | 73.0 | 73.1 | 72.9 | / |
> | 10 | 72.5 | 72.2 | 72.6 | 70.9 |
>
> The head row is the $lr_{structure}$, while the head column is the $\lambda_{resource}$. '/' indicates that the model cannot reach our target resource constraint.
> Obviously,
>
> 1. Smaller $\lambda_{resource}$ is better, as long as the model can reach the target resource constraint. Smaller $\lambda_{resource}$ means that the task loss takes more control of the update of the structure parameter.
> 2. When $\lambda_{resource}$ is small, the model is not sensitive to the change of $lr_{structure}$. When $\lambda_{resource}$ is large, a relatively large $lr_{structure}$ is better. This is because reaching the target resource constraint can reduce the influence of the resource constraint loss, as resource constraint loss is zero when the model reaches the target resource constraint.
>
> Therefore, the setting of $\lambda_{resource}$ and $lr_{structure}$ is not difficult. We first fix $lr_{structure}$ and turn $\lambda_{resource}$ to a small value and ensure the model can reach the target resource constraint. Then, we turn $lr_{structure}$ to a relatively large value, which makes the model reach the target resource constraint in the first hundreds of iterations. Only observing the resource decrease in the first epoch is enough to set these two hyperparameters.
>
> Compared with other NAS methods, our method uses fewer hyperparameters. For example, ModelAmplification must turn at least five hyperparameters for different tasks and models.
>
> We added the ablation study to Appendix A.5.4 in our revised paper.
>
> [1] Molchanov, Pavlo, et al. "Importance estimation for neural network pruning." Proceedings of the IEEE/CVF conference on computer vision and pattern recognition. 2019.
> [2] Humble, Ryan, et al. "Soft masking for cost-constrained channel pruning." European Conference on Computer Vision. Cham: Springer Nature Switzerland, 2022.
> [3] Neuron-level Structured Pruning using Polarization Regularizer

---

> ### Author Response · Authors · 2023-11-16
> **Response to Reviewer 9aAa (Part 3/3)**
>
> > W5.1 "The paper's ablation study is not comprehensive enough to demonstrate the advantages of the proposed method. The paper only compares the differential topk with a uniform scaling baseline, but does not compare it with other model scaling methods, such as compound scaling or progressive scaling."
>
> Thanks for your suggestion. We don't compare with compound scaling and progressive scaling in our ablation study because we have already compared them with our method in our main experiments. Where EfficientNet is a compound scaling method, and ModelAmplification is a progressive scaling method. Besides, we have also compared our method with other model scaling methods, such as differential scaling (JointPruning).
> Therefore, we think it is not necessary to compare with other scaling methods in our ablation study.
>
> > W5.2 " The paper also does not show how the differential topk performs on different network layers, such as convolutional layers or attention layers."
>
> We apologize for our unclear description.
> We apply our differential topk to different layers by multiplying the mask, outputed by our differential topk operators, with the input to the layer.
>
> For convolutional layers, suppose the input is $X \in \mathbb{R}^{B \times C \times H \times W}$, and the mask is reshaped as $m \in \mathbb{R}^{1 \times C \times 1 \times 1}$, $X \times m$ works as the new input to the layer.
>
> For an attention layer, we search the head dims of q k v and the number of heads. Suppose our supernet has $H$ heads and $D$ dims in each head. We have a mask for qk head dim with $m_{qk}\in R^{1\times 1 \times 1 \times D}$, a mask for v head dim with $m_{v}\in R^{1\times 1 \times 1 \times D}$, and a mask for number of heads $m_{head}\in R^{1\times H \times 1 \times 1}$. Suppose the sequence length is $L$, and the q k v for self-attention is $Q,K,V \in R^{B \times H \times L \times D}$. We compute the output of the self-attention by $softmax(\frac{Q'K'^T}{\sqrt{}})V'$, where $Q'=Q\times m_{qk}\times m_{head}, K'=K\times m_{qk}\times m_{head}, V'=V\times m_{v}\times m_{head}$
>
> We added above content to Appendix A.1.1
>
> > Q1 "In Section 3.1.3, you use a moving average to update the layerwise mask. What is the motivation and benefit of this technique? How does it affect the convergence and stability of the optimization?"
>
> The moving average is used to make element importance measures more stable. As shown in the following table.
> With more stable element importance, we can get a better result.
> | Measure Method | Top-1 |
> |----------------|-------|
> | Taylor without moving average | 72.5 |
> | Taylor with moving average | **73.1** |
>
> Taylor importance with moving average is a common technique in the pruning field. Please refer to [1,2] for more details.
> We also added the ablation study to Appendix A.5.3 in our revised paper.
>
>
> [1] Molchanov, Pavlo, et al. "Importance estimation for neural network pruning." Proceedings of the IEEE/CVF conference on computer vision and pattern recognition. 2019.
> [2] Humble, Ryan, et al. "Soft masking for cost-constrained channel pruning." European Conference on Computer Vision. Cham: Springer Nature Switzerland, 2022.
>
> > Q2 "In Section 3.1.3, you adopt an L1-norm regularization term for the channel-wise mask. Why did you choose this norm over other alternatives, such as L2-norm or entropy? How does the choice of norm influence the sparsity and diversity of the channel-wise mask?"
>
> Sorry for the confusion. We did not use L1-norm regularization term in the paper. We mentioned that "L1-norm" can be an alternative as an importance measure method. The "L1-norm" refers to a well-known channel importance measure method, which computes the L1-norm of each filter as the importance score. Please refer to [1] for more details.
>
> [1] Li, Hao, et al. "Pruning filters for efficient convnets." arXiv preprint arXiv:1608.08710 (2016).
>
> Thanks for diving into our paper and providing the constructive comments. We have revised our paper according to your suggestions. Please feel free to contact us if you have any further questions.
> Furthermore, we want to emphasize the value of our work. Our method. It's a general and flexible NAS method that can be applied in real-world scenarios. It improves performance stably, has only two hyperparameters that need turning, and costs much fewer resources than other NAS methods. Hence, we think our work is valuable and deserves to be published.
> If our responses have addressed your concerns, we would appreciate it if you could consider raising the score of our paper. Thank you for your time and consideration.

---

> > ### Comment · Reviewer_9aAa · 2023-11-20
> > **About Q1 & Q2**
> >
> > I noticed that you didn't directly address the question at hand. In the study titled 'Zero-cost Proxies for Efficient NAS,' the authors explore various pruning-based metrics used to assess the importance of neurons, each exhibiting distinct performance characteristics. Some of these proxies include:
> >
> > Plain: $S(\theta) = \frac{\partial \mathcal{L}}{\partial \theta} \circ \theta$
> > SNIP: $S(\theta) = \left| \frac{\partial \mathcal{L}}{\partial \theta} \circ \theta \right|$
> > GRASP: $S(\theta) = -\left( H \frac{\partial \mathcal{L}}{\partial \theta} \right) \circ \theta$
> > Fisher: $S(z) = \sum_{z_i \in Z} \left( \frac{\partial \mathcal{L}}{\partial z_i} \right)^2$
> > SynFlow: $R = \prod_{\theta_i \in \Theta} \left| \theta_i \right|$, $S(\theta) = \left( \frac{\partial R}{\partial \theta} \right) \circ \theta$
> >
> > Among these proxies, Taylor (Plain) is similar to Plain. My question is why you chose Taylor (Plain) instead of other proxies such as SynFlow, which are considered to be more accurate.

---

> > > ### Author Response · Authors · 2023-11-22
> > > **Additional response to "About Q1 and Q2"**
> > >
> > > Here, we compare performance with different importance metrics, including Taylor(ours), SNIP, and Fisher importance. We exclude GRASP and SynFlow. The reasons are listed below.
> > >
> > > 1. GRASP, $S_i=-H(\frac{\partial L}{\partial m})m_i$, needs to compute hessian $H$, by several forward and backward passes each time. It's not suitable for our method, as we train our model and structure parameters during the search. It will cost too much time to compute hessian each iteration.
> > > 2. SynFlow, $S_{i}^{l}=[\mathbb{1}^T\prod_{k=l+1}^N|m^k|]\;\;|m^l_i|\;\;[\prod_{k=1}^{l-1} |m^k|\mathbb{1}]$, is proposed to address the problem of comparing importance from different layers. While we only need to compare the importance in the same layer, SynFlow is equivalent to $S_i=m_i$, which never changes. Therefore, SynFlow is also not suitable for our method.
> > >
> > > We modify SNIP and Fisher importance to make them suitable for our method. The results are shown in the following table.
> > >
> > > | Metric | Defination                               | Top-1 |
> > > | ------ | ---------------------------------------- | ----- |
> > > | Taylor | $(\frac{\partial L}{\partial m_i}m_i)^2$ | 73.1  |
> > > | SNIP   | $\|\frac{\partial L}{\partial m_i}m_i\|$ | 73.0  |
> > > | Fisher | $(\frac{\partial L}{\partial m_i})^2$    | 73.2  |
> > >
> > > The difference between different metrics is not significant. Our results support the conclusion of [1].
> > > We think these results are reasonable, as they are very similar to each other.
> > >
> > > Besides, there are two reasons why ZiCo finds the difference between different metrics is significant.
> > > 1. The search granularity is different. ZiCo searches models, while we search structure elements (channels/layers). Larger search granularity may make the metric more important.
> > > 2. The search method is different. ZiCo uses a one-shot search method without any training, while we train our model and structure parameters during the search. The training process makes the metric less important for our method.
> > >
> > > Now we have completed our response to your additional questions. If you have any further questions, please let us know.
> > > If our responses have addressed your concerns, we would appreciate it if you could consider raising the score of our paper. Thank you very much.
> > >
> > > [1] Huang, Zhongzhan, et al. "Rethinking the pruning criteria for convolutional neural network." Advances in Neural Information Processing Systems 34 (2021): 16305-16318.

---

> > > > ### Comment · Reviewer_9aAa · 2023-11-22
> > > > **Increase Rating**
> > > >
> > > > Sorry for the late reply. I appreciate your efforts in responding to my questions. I have decided to increase the rating.

---

> > > > > ### Author Response · Authors · 2023-11-22
> > > > > **Thanks for Increasing the Rating**
> > > > >
> > > > > Dear reviewer 9aAa:
> > > > >
> > > > > We sincerely appreciate your recognition of our work.
> > > > >
> > > > > Thanks a lot, and best wishes,
> > > > >
> > > > > Authors

---

> ### Author Response · Authors · 2023-11-20
> **Responds to "About Q1 & Q2"**
>
> Sorry, we misunderstood your question.
>
> In our view, your question can be divided into two parts.
> 1. Why do we use Taylor importance with moving average as the element importance measure method?
>
> Anwser: Our reason is simple. Our main contribution is to propose a method to search the number of elements, rather than measuring the importance of each element. Hence, we directly select a **simple and commonly used** element importance metric, Taylor importance, as the element importance measure method.
>
> 2. What's the advantage of Taylor importance with moving average, compared with other metrics?
>
> Anwser:
>
> To be honest, we do not conduct a comprehensive comparison between different metrics, as this is not our main contribution. As a comparison, ZiCo must compare with other metrics, as they propose a new metric.
>
> Besides, For structural pruning, some work argues that the difference between different importance metrics is not significant [1]. Note the conclusion of Zico about the importance metrics is not suitable for us, as the search granularity is different. ZiCo modifies the importance metrics to select models, while we search structure elements (channels/layers).
>
> Although we do not conduct a comprehensive comparison, we will still provide an ablation study to compare different metrics.
>
> [1] Huang, Zhongzhan, et al. "Rethinking the pruning criteria for convolutional neural network." Advances in Neural Information Processing Systems 34 (2021): 16305-16318.

---

### Official Review · Reviewer_EkDu · 2023-10-31

**Soundness:** 3 good
**Presentation:** 3 good
**Contribution:** 3 good
**Rating:** 6
**Confidence:** 5

**Summary:**

The paper discusses the challenges of manually designing network architectures and the nondifferentiable limitations of existing Neural Architecture Search (NAS) methods. To address these issues, the authors propose Differential Model Scaling (DMS), which offers increased efficiency in searching for optimal width and depth configurations in DNNs. DMS allows for direct and fully differentiable modeling of both width and depth, making it easy to optimize. The authors evaluate DMS across various tasks, including image classification, object detection, and language modeling, using different network architectures such as CNNs and Transformers. The results consistently demonstrate that DMS can find improved network structures and outperforms state-of-the-art NAS methods.

**Strengths:**

1. This paper proposes a differentiable top-k method, which could be used to select channels or layers in DNNs. The design of differentiable top-k  method is skillful and  meaningful. With normalized importance factors, a learnable parameter $\alpha$ is used to select elements.
2. The whole DMS method merged the task loss and cost loss, With the guidence of cost loss, the DMS can search for efficient models.
3. Various experiments demonstrates the superiority of DMS over existing NAS methods. The pruning experiments  presents the method is better than SOTA pruning methods.

**Weaknesses:**

1. Different element importance methods are not studied. Some comparisions should be presented to underscore the DMS method.
2. More types of cost losses should be considered, such as latency or memory cost. Latency is a superior indicator compare to FLOPs.
3. As far as I know, gumbel top-k method is also differentiable, why you develop a new differentiable top-k methd?
4. The open source of the code will help to understand the paper.

**Questions:**

Please see the weaknesses.
Besides, in p.5, the authors demonstrate that "Intuitively, $c_{i}^{'}$ indicates the portion of $c$ values larger than $c_{i}$.". Here should be "smaller" instead of "larger".

---

> ### Author Response · Authors · 2023-11-16
> **Response to Reviewer EkDu**
>
> Dear review:
>
> Thanks for your constructive comments. We address your comments as follows:
>
> > W1: "Different element importance methods are not studied. Some comparisons should be presented to underscore the DMS method."
>
> Thanks for your suggestion. Here, we provide a new ablation study to show the influence of element importance.
>
> | Measure Method | Top-1 |
> |----------------|-------|
> | Index | 72.3 |
> | Taylor, without moving average | 72.5 |
> | Taylor, with moving average | **73.1** |
>
> "Index" means assigning an importance value to each element according to its index in the sequence statically. For example, we assign 0 to the first element, and 1.0 to the last.
>
> Obversely, Taylor importance works better than index importance, as it is able to detect the importance of each element dynamically. Moving averaging further makes Taylor importance more stable.
>
> We also added the ablation study to Appendix A.5.3 in our revised paper.
>
> > W2: "More types of cost losses should be considered, such as latency or memory cost. Latency is a superior indicator compare to FLOPs."
>
> Thanks for your suggestion. In this paper, we use two different resource constraint losses, including Flops loss and number of parameters loss.
>
> In this paper, our main target is to show the high search efficiency of our method rather than design a model for a specific platform. Therefore, we think any resource indicator is acceptable as long as we use the same resource indicator as our baselines for fair comparison.
>
> Specifically, for memory cost constraint, we think it's similar to the number of parameters. Hence, we think there is no need to add a memory cost constraint to our paper.
>
> For latency constraint, We agree that latency is a useful indicator. However, the relation between latency and structure hyperparameters is not easy to model. It means latency is not easy to be converted to a differentiable loss. We think it's also an interesting research direction. We will take it in future research.
>
> Therefore, we think the constraints on flops and the number of parameters are enough to evaluate our method, and other constraints should be left for future research, such as some work that focuses on a specific hardware platform.
>
>
> > W3: "As far as I know, gumbel top-k method is also differentiable, why you develop a new differentiable top-k methd?"
>
> We are sorry for the confusion.
> Gumbel top-k [1,2] is completely different from our method. It may be confusing because we use the same term, "top-k", in our paper.
>
> Gumbel top-k is learned to sample 'k' different elements from a set of n elements, where 'k' is a constant.
> Our differential topk is learning the 'k' itself, which is a learnable parameter.
> Gumbel top-k cannot be used to search the width and depth of a network, while our method can.
>
> We also talk about another method, "Gumbel softmax", which is also usually used in pruning and NAS [3].
> "Gumbel softmax" is able to work as a learnable gate to select elements for width and depth. However, as we described in our paper, it models depth and width hyperparameter searching as an element selection problem, making it hard to optimize.
>
> [1] Tan, Haoxian, et al. "Mutually-aware Sub-Graphs Differentiable Architecture Search." arXiv preprint arXiv:2107.04324 (2021).
> [2] Li, Jiaoda, Ryan Cotterell, and Mrinmaya Sachan. "Differentiable subset pruning of transformer heads." Transactions of the Association for Computational Linguistics 9 (2021): 1442-1459.
> [3] Herrmann, Charles, Richard Strong Bowen, and Ramin Zabih. "Channel selection using gumbel softmax." European Conference on Computer Vision. Cham: Springer International Publishing, 2020.
>
>
> > W4: "The open source of the code will help to understand the paper."
>
> Considering the double-blind review policy, we can not open-source our code before the paper is accepted. We will open-source our code after the paper is accepted.
>
> > Q5: "Besides, in p.5, the authors demonstrate that "Intuitively, $c_i'$ indicates the portion of $c$ values larger than $c_i$ Here should be "smaller" instead of "larger".
>
> Thanks for your correction. We have updated the paper to correct it.
>
> Thanks for your effort in reviewing our paper. We have addressed all your comments in above. If you have any further questions, please let us know.

---

> ### Author Response · Authors · 2023-11-21
> **Sincerely expecting feedback from reviewer EkDu.**
>
> Dear reviewer:
>
> Thanks for your constructive comments. We have posted our responses to your comments. We expect your feedback about whether our responses address your concerns, or if you have any further questions. We are glad to answer them and improve our paper.
>
> Best,
>
> Authors

---

### Official Review · Reviewer_nwwR · 2023-10-31

**Soundness:** 2 fair
**Presentation:** 2 fair
**Contribution:** 2 fair
**Rating:** 3
**Confidence:** 2

**Summary:**

This paper proposes a new method for differentiable architecture search using a new differentiable top-k operator. Elements (units, blocks, filters, any grouping of parameters) of the network are assigned importance parameters, $c$, that depend on a moving average of the [Taylor importance][taylor]. A learnable threshold $a$ is used to select $k$ elements whose $c$ exceed the threshold. A tradeoff in performance versus capacity is achieved by computing the model resource usage from $c$ and $a$ then constructing a loss function:
$$
 \text{loss}_{\text{resource}} = \log \frac{r_c}{r_t} \text{ if } r_c > r_t \text{ else } 0
$$
where $r_c$ is the current resource consumption and $r_t$ is the target.

During stochastic gradient optimization, $a$ will be pushed to maintain resource consumption at the desired level, while providing some slack for the model to still learn to perform the task.

The effectiveness of this method is tested in experiments on image classification on ImageNet, image detection on COCO and a large language model finetuning task.

[taylor]: https://arxiv.org/abs/1906.10771

**Strengths:**

The main contributions of the paper are the empirical results: outperforming [ZiCo][] by 0.7% with the same search time (0.4 GPU days).
This result appears to be well tested and therefore the paper achieves this goal. Similar results also support the method empirically on COCO and language model finetuning.

The authors describe the key difference between this work and similar architecture search methods is that it provides a differentiable and direct way to approach architecture search. In other words, other works allow a differentiable selection of which elements to include but do not allow easy optimization of how many elements to include.

Architecture search is a significant area of research and this paper submits a new method for consideration.

[zico]: https://arxiv.org/abs/2301.11300

**Weaknesses:**

In Section 3.2 the authors mention that this method bears some resemblance to pruning works, "Our DMS follows a pipeline similar to training-based model pruning." This implies that the model should be compared to pruning based methods in experiments. However, the comparisons appear to be made to NAS methods, such as [JointPruning][]. A comparison to state of the art sparse methods, such as [RIGL][] would make the experiments more robust.

The function they have constructed for optimization is smooth but saturates outside of the active regions illustrated in Figure 2. This may cause vanishing gradient information. Any experiment to investigate whether this happens during training, or why it doesn't happen would be valuable.

The relationship between resource constraint and the top-k parameters is in the appendix but it's extremely important to the overall algorithm.

The top-k operator as described leads the reader to assume $k$ would be fixed but in practice it's not constrained and $k$ can be any value. Really it's just a binary mask that has a soft constraint to sum to a low enough value to meet the resource constraints.

[rigl]: https://arxiv.org/abs/1911.11134
[jointpruning]: https://ieeexplore.ieee.org/document/9516010

**Questions:**

How sensitive is the method to the element importance measure $c$? It's computed as a moving average with a specific hyperparameter. It seems like the gradient estimate of $a$ depends on this being stable.

In Section 3.2 you say "Compared with training-based pruning, our method eliminates the need for time-consuming pretraining since we think searching from scratch is more efficient thatn from a pretrained model...". How does that save resources? Typically pretrained models are available for free, but training one yourself from scratch is extremely expensive? I don't understand what Table 4 means because the rows and colums refer to search and retraining but one can't have both a pretrained model and a model that is retrained?

The gains over prior architecture search methods seem to be relatively minor, such as 0.7% accuracy on ImageNet. What would be your argument against this?

---

> ### Author Response · Authors · 2023-11-16
> **Response to Reviewer nwwR (Part 1/3)**
>
> Dear reviewer:
>
> Thanks for your helpful comments. Here are our responses to your comments.
>
> > W1: "In Section 3.2 the authors mention that this method bears some resemblance to pruning works, "Our DMS follows a pipeline similar to training-based model pruning." This implies that the model should be compared to pruning based methods in experiments. However, the comparisons appear to be made to NAS methods, such as JointPruning. A comparison to state of the art sparse methods, such as RIGL would make the experiments more robust."
>
> Thanks for your suggestion. We compared our method with SOTA structure pruning methods in our original paper, detailed in Appendix A.4. Our method also outperforms SOTA structure pruning methods.
>
> There are two types of pruning methods: unstructural pruning and structural pruning. Unstructural pruning methods, such as RIGL, prune the model by setting the weights to zero. Structural pruning methods, such as our method, prune the model by removing the whole dimensions of the weight tensor. The two types of pruning methods have different pruning granularity. Unstructural pruning needs special hardware support to achieve acceleration, while structural pruning can be accelerated by general hardware. Therefore, RIGL can not be compared with our method directly. Please refer to [1] for more details.
>
>
> Besides, we want to talk about the boundary between structure pruning and NAS. In fact, they are not two independent fields. The classic NAS method, darts [2], is very similar to pruning, and the pruning work, EagleEye[3], also searches model architectures.
> Our method can be interpreted as a structure pruning method and a NAS method simultaneously. Propose it as a nas method because we want to emphasize our method's high structure search efficiency rather than discovering better initialization from pretrained models.
>
> [1] Li, Hao, et al. "Pruning filters for efficient convnets." arXiv preprint arXiv:1608.08710 (2016).
> [2] Liu, Hanxiao, Karen Simonyan, and Yiming Yang. "Darts: Differentiable architecture search." arXiv preprint arXiv:1806.09055 (2018).
> [3] Li, Bailin, et al. "Eagleeye: Fast sub-net evaluation for efficient neural network pruning." Computer Vision–ECCV 2020: 16th European Conference, Glasgow, UK, August 23–28, 2020, Proceedings, Part II 16. Springer International Publishing, 2020.
>
>
> > W2: "The function they have constructed for optimization is smooth but saturates outside of the active regions illustrated in Figure 2. This may cause vanishing gradient information. Any experiment to investigate whether this happens during training, or why it doesn't happen would be valuable."
>
> Through our deduction and observation, we think gradient vanishing does not happen in our method. We show our deduction as follows.
>
> Figure 2 illustrates the gradient of $a$ with respect to element mask $m_i$. Here, we denote it as $\frac{\partial m_i}{\partial a}$. It's indeed that $\frac{\partial m_i}{\partial a}$ may equal to 0 when $m_i$ is close to 0 or 1. But the gradient of $a$ with respect to the $task\_loss$ is $\sum_{i=0}^{N}{\frac{\partial task \_loss}{\partial m_i}\frac{\partial m_i}{\partial a}}$. As shown in Figure 2, $\frac{\partial m_i}{\partial a}$ will not be 0 for all $i \in [0,N]$. Therefore, the gradient of $a$ has a low probability of being 0.
>
> Besides, we also do not observe gradient vanishing in our experiments. Therefore, we think gradient vanishing does not happen in our method.
>
>
> > W3: "The relationship between resource constraint and the top-k parameters is in the appendix but it's extremely important to the overall algorithm."
>
> We apologize for the inconvenience. As many papers have used the resource constraint[1,2], we move the section about the resource constraint to the appendix when the paper is too long.
>
> Thanks for your advice. We will consider moving the section to the main paper.
>
> [1] Li, Yanyu, et al. "Pruning-as-Search: Efficient Neural Architecture Search via Channel Pruning and Structural Reparameterization."
> [2] Gao, Shangqian, et al. "Disentangled differentiable network pruning." European Conference on Computer Vision. Cham: Springer Nature Switzerland, 2022.
>
>
> > W4: "The top-k operator as described leads the reader to assume would be fixed but in practice it's not constrained and can be any value. Really it's just a binary mask that has a soft constraint to sum to a low enough value to meet the resource constraints."
>
> We are sorry for the confusion. We agree that the 'k' is usually fixed.
> The reason we use the term "top-k" is because we just follow prior work. For example, [1] uses "soft top-k" in their method with an unfixed 'k'.
> We will consider more carefully when using the term "top-k" in the future.
>
> [1] Gao, Shangqian, et al. "Disentangled differentiable network pruning." European Conference on Computer Vision. Cham: Springer Nature Switzerland, 2022.

---

> ### Author Response · Authors · 2023-11-16
> **Response to Reviewer nwwR (Part 2/3)**
>
> > Q1: "How sensitive is the method to the element importance measure $a$? It's computed as a moving average with a specific hyperparameter. It seems like the gradient estimate of $a$ depends on this being stable."
>
> Yes, the moving average is used to make the training more stable.
> Here, we provide an ablation study of the element importance measure.
>
> | Measure Method | Top-1 |
> |----------------|-------|
> | Index | 72.3 |
> | Taylor importance without moving average | 72.5 |
> | Taylor importance with moving average | 73.1 |
>
> "Index" means assigning an importance value to each element according to its index in the sequence statically. For example, we assign 0 to the first element, 1.0 to the last.
> As shown in the above Table, the Taylor importance and the moving average improve the performance.
> As Taylor importance computes importance by mini-batch, the moving average is used to make the importance measure more stable, also making the gradient computation of $a$ more stable.
>
> We also added the ablation study to our revised paper in Appendix A.5.3.
>
>
> > Q6: "In Section 3.2 you say "Compared with training-based pruning, our method eliminates the need for time-consuming pretraining since we think searching from scratch is more efficient thatn from a pretrained model...". How does that save resources? Typically pretrained models are available for free, but training one yourself from scratch is extremely expensive?"
>
> Here, we compare the pipeline of our method with standard pruning methods as follows.
> | Method | Pruning | Ours |
> |--------|---------|------|
> | Stage 1 | Pretrain | No Need |
> | Stage 2 | Prune | Search |
> | Stage 3 | Fintune | Retrain |
>
> We guess that you may think the fintune stage of pruning methods is more efficient than the Retrain stage (train from scratch) of our method. However, most structural pruning methods directly use the same training setting as pretraining in fintuning stage [1,2]. Because this setting achieves the best performance.
> Therefore, our method is more efficient by omitting the most resource-consuming pretraining stage and does not introduce any extra cost.
>
> Besides, the pretrained models are not free but expensive. They are free only because someone has paid for them and shared them with us.
> Imagine two situations:
>
> 1. We want to search for a model with a big private dataset.
> 2. We want to design a new model architecture.
>
> In both situations, we can not find any free pretrained model.
>
> Here, we provide the performance and search cost comparison between searching with pretrained models and without pretrained models.
>
> | Supernet | $Iinit_{search}$ | $cost_{pretrain}$ | $cost_{search}$ | $cost_{total}$ | Top-1 |
> |----------|------------------|-------------------|-----------------|----------------|-------|
> | ResNet-50 | Random | 0 | 41 | 41 | 73.1 |
> | ResNet-50 | Pretrain | 410 | 41 | 451 | 73.8 |
> | ResNet-152 | Random | 0 | 116 | 116 | **74.6** |
>
> (The unit of cost is $G MACs \times Epochs$)
> Searching without pretrained models achieves 0.8\% higher top-1 accuracy and only uses nearly 1/4 of the total cost of searching with pretrained models.
> As a result, we think our method is more efficient than standard pruning methods in both performance and cost.
>
> [1] Li, Bailin, et al. "Eagleeye: Fast sub-net evaluation for efficient neural network pruning." Computer Vision–ECCV 2020: 16th European Conference, Glasgow, UK, August 23–28, 2020, Proceedings, Part II 16. Springer International Publishing, 2020.
> [2] Wang, Huan, and Yun Fu. "Trainability preserving neural structured pruning." arXiv preprint arXiv:2207.12534 (2022).
>
> > Q7 "I don't understand what Table 4 means because the rows and columns refer to search and retraining, but one can't have both a pretrained model and a model that is retrained?"
>
> We apologize for the confusion. We update the table as follows:
> | Supernet | $Iinit_{search}$ | $Init_{retrain}$ | $cost_{pretrain}$ | $cost_{search}$ | $cost_{total}$ | Top-1 |
> |----------|------------------|------------------|-------------------|-----------------|----------------|-------|
> | ResNet-50 | Random | Random | 0 | 41 | 41 | 72.6 |
> | ResNet-50 | Pretrained | Random | 410 | 41 | 451 | 72.5 |
> | ResNet-50 | Random | Searched | 0 | 41 | 41 | 73.1 |
> | ResNet-50 | Pretrained | Searched | 410 | 41 | 451 | 73.8 |
>
> This table shows different initialization schemes for the search stage and retrain stage. The search stage searches a structure, and then the structure is retrained. Hence, in the retrain stage, the model is initialized randomly or with the weights from the search stage. So there is no pretrained initialization in the retrain stage.

---

> ### Author Response · Authors · 2023-11-16
> **Response to Reviewer nwwR (Part 3/3)**
>
> > Q8 "The gains over prior architecture search methods seem to be relatively minor, such as 0.7% accuracy on ImageNet. What would be your argument against this?"
>
> There are two reasons.
> 1. The comparison is unfair, as Zico uses a stronger training script, such as distillation and mix-up. Therefore, the real improvement should be larger.
> 2.  Our main target is not to provide SOTA ImageNet performance but to provide a general and flexible architecture search method.  Our method is almost compatible with most network architectures and training processes. Therefore, we only change the depth and width of our baselines and do not use any tricks to improve performance. For our method, improving various tasks stably with limited search costs and limited hyperparameter tuning is more critical, as it is more important for real-world applications. Note, as far as we know, we are the first method that achieves 2.0% performance improvement on yolo-v8, a high-performance and lightweight model.
>
> Therefore, we think 0.7% improvement is not minor.
>
>
> Thank you for your comments again. We are looking forward to hearing more feedback from you.
> If you have any further questions, please feel free to contact us. We are active until the rebuttal deadline.
> Besides, we want to emphasize the value of our work. Our method is novel and effective, and it outperforms the state-of-the-art NAS methods using much fewer resources. It pushes the boundary of NAS and makes it more practical. Our work is valuable and deserves to be introduced to the community.
> Would you consider raising the score if our responses have resolved your concerns? Thank you!

---

> ### Author Response · Authors · 2023-11-21
> **Sincerely expecting feedback from reviewer nwwR.**
>
> Dear reviewer:
>
> Thanks for your constructive comments. We have posted our responses to your comments. We expect your feedback about whether our responses address your concerns, or if you have any further questions. We are glad to answer them and improve our paper.
>
> Best,
>
> Authors

---

> ### Comment · Reviewer_nwwR · 2023-11-21
> **Rebuttal Response**
>
> I apologise for not replying sooner.
>
> W1: I missed Appendix A.4 when reading this paper for the first time. It does appear that the method compares well in these experiments.
>
> W2: Figure 2 does show regions of zero gradient. Other methods that use this kind of relaxation add noise (most famously Gumbel softmax) to smooth out the gradient (with noise sometimes it will land in the high gradient region). This appears to involve a fixed mask, perhaps the noise from SGD is enough to maintain gradient information. If, as you say, "we also do not observe gradient vanishing in our experiments" then it may be worthwhile adding those results, for example a gradient hook showing the norm of the gradient passing through the soft mask versus the gradient norm due to the resource loss would be valuable. It seems like the method requires these two parts to be balanced carefully via $\lambda_{resource}$.
>
> Q1: I find it concernging that the Taylor importance is not critical to the performance of the method. Why would it still work within 1% accuracy without it? The importance signal tells the model which parts are important. Is the method in this analogous to [Nested Dropout](https://proceedings.mlr.press/v32/rippel14.html), ie the network reacts to the importance of different elements prescribed by the mask and not the other way around?
>
> Q6: I don't understand how you can end up with a trained model here without training it; you're comparing pretrained models to your search method and the cost is 116 for the larger model using your search method? And it performs better? I apologize I don't understand what is happening here and I will update my confidence in my review.
>
> Q7: See response to response Q6 above.
>
> Q8: Unfortunately, I can't conclude that the method would have worked better if it had been trained differently. It would be necessary to replicate ZiCo and train it without the extra tricks they used for a fair comparison.

---

> > ### Author Response · Authors · 2023-11-22
> > **Response to "Rebuttal Response" Part 1/2**
> >
> > Thanks for your feedback. Here we address your concerns below.
> >
> > > W2: Figure 2 ...$\lambda_{resource}$
> >
> >
> >
> > ## 1. $g_a$ decides whether the gradient vanishing problem exists, rather than $\frac{\partial m_i}{\partial a}$.
> >
> > Here, we think it's important to distinguish two situations, where some gradients are close to zero.
> >
> > 1. The **total gradient** of a learnable weight is close to zero. It means the weight is not updated, and causes the gradient vanishing problem.
> > 2. The **partial gradient** of a learnable weight is close to zero. The weight may receive gradients from other parts, making its total gradient not close to zero. Therefore, it does **not** cause the gradient vanishing problem.
> >
> > Back to our method. Let's first define the gradient of a learnable structure parameter $a$ as below. Here, we ignore the gradient from $L_{resource}$, as it's never close to zero when $L_{resource}!=0$.
> >
> > $g_{a}=\frac{\partial L_{task}}{\partial a}=\sum_{i=0}^{N}{\frac{\partial L_{task}}{\partial m_i}\frac{\partial m_i}{\partial a}}$
> >
> > $g_a$ is the total gradient of $a$, and $\frac{\partial m_i}{\partial a}$ is a partial gradient of $a$. Therefore, it's $g_a$ that decides whether the gradient vanishing problem exists, rather than $\frac{\partial m_i}{\partial a}$. Note Figure 2 illustrates $\frac{\partial m_i}{\partial a}$, not $g_a$.
> >
> > ## 2. Why $g_{a}$ is not equal to 0.
> >
> > We define $m_i=\text{Sigmoid}(\lambda(c'_i-a)), i\in [0,N), \lambda=N$ in our paper. As $c'_i, i\in [0,N)$ distributes evenly in the range of $[0,1]$, we can directly assign $c'_i=i/N$.
> >
> > There are two cases of $\frac{\partial m_i}{\partial a}$
> >
> >
> > 1. $|\frac{\partial m_i}{\partial a}| <0.05\lambda\approx 0,\; \text{if} |i-aN|>3$
> > 2. $|\frac{\partial m_i}{\partial a}| \in [0.05\lambda,0.25\lambda],\; \text{if} |i-aN|\leq3$
> >
> > There are at least six cases for $i$ making $\frac{\partial m_i}{\partial a}$ not close to zero.
> > Besides, as $\frac{\partial L_{task}}{\partial m_i}$ has low probability to be close to zero, $g_a=\sum_{i=0}^{N}{\frac{\partial L_{task}}{\partial m_i}\frac{\partial m_i}{\partial a}}$ is not close to zero.
> > Therefore, it's reasonable that we do not observe gradient vanishing in our experiments.
> >
> > ## 3. Compare with Other Methods, like Gumbel softmax
> >
> > We method is different from gumbel-softmax-like methods.
> >
> > 1. Gumbel-softmax-like methods use one parameter for one element. They generate mask $m_i=f(b_i)$. Therefore the structure parameters is $b=\{b_i\}^N \in \mathbb{R}^N$ for one layer.
> > 2. We use one parameter for all elements in one layer. We generate mask $m_i=f(a,c_i')$, note $c_i'$ is not a parameter. Therefore, the structure parameter is $a \in \mathbb{R}$ for one layer.
> >
> > We argue there are two reasons why gumbel-softmax-like methods add relaxation
> >
> > 1. Their structure parameters are too many to train. The relaxation is a regularization on the structure parameters to make them easier to train.
> > 2. They need to polarize the mask $m_i$ to 0 or 1. The relaxation is a trick to make the mask $m_i$ more polarized.
> >
> > Gradient vanishing is usually caused by the above two reasons.
> >
> > Therefore, there are two reasons why we do not need to add relaxation and there is no gradient vanishing problem in our method.
> >
> > 1. The number of our structure parameters is much fewer than gumbel-softmax-like methods. The gradient information is gathered to one parameter, rather than distributed to $N$ parameters, making it easier to train.
> > 2. We use one parameter to generate mask $m_i$, which achieves much better polarization than gumbel-softmax-like methods, we do not need to add any other relaxation.
> >
> > Therefore, we think the need for relaxation of gumbel-softmax-like methods can not used to judge whether there is a gradient vanishing problem in our method. Conversely, their need for relaxation proves the privilege of our method: **dircetly and differentable modeling the structure parameters, which is much more easier to optimize**.
> >
> > We also print the gradients of $a$ from $L_{task}$ and ${L_{resource}}$ respectively in our experiments. They are not close to zero.
> >
> > In conclusion, we make sure there is no gradient vanishing problem in our method.

---

> > ### Author Response · Authors · 2023-11-22
> > **Response to "Rebuttal Response" Part 2/2**
> >
> > > Q1: I find it concernging that the Taylor importance ...?
> >
> >
> > We think there are two aspects of why it still works when we do not use taylor importance.
> >
> > 1. We agree with you that the method in this situation is similar to Nested Dropout, which is also a way to find a set of weights, better than random initialization, for the model. Index importance is also widely used in NAS methods, such as [1]. Their experiments also support this opinion.
> > 2. There are mainly two aspects of why pruning/NAS methods improve the performance, including finding better initialization for the model and finding a better structure for the model[2,3]. As index importance is also able to search better structure for the model, the method still works (improves the performance of the model).
> >
> > [1] Yu, Jiahui, and Thomas Huang. "Autoslim: Towards one-shot architecture search for channel numbers." arXiv preprint arXiv:1903.11728 (2019).
> > [2] Frankle, Jonathan, and Michael Carbin. "The lottery ticket hypothesis: Finding sparse, trainable neural networks." arXiv preprint arXiv:1803.03635 (2018).
> > [3] Liu, Zhuang, et al. "Rethinking the value of network pruning." arXiv preprint arXiv:1810.05270 (2018).
> >
> > > Q6 and Q7:  I don't understand how you can end up with a trained model here without training it ...
> >
> > We provide a more detailed table below. In this table, we prune the supernets to models with 1G flops. "N" indicates we do not pretrain the supernet, and "Y" indicates we pretrain the supernet.
> >
> >
> > | Supernet   | Pretrain Stage | Search Stage               | Retrain Stage            | $cost_{pretrain}$ | $cost_{search}$ | $Cost_{retrain}$ | $cost_{total}$ | Top-1    |
> > | ---------- | -------------- | -------------------------- | ------------------------ | ----------------- | --------------- | ---------------- | -------------- | -------- |
> > | ResNet-50  | N              | initialize randomly        | initialize randomly      | 0                 | 41              | 100              | 141            | 72.6     |
> > | ResNet-50  | Y              | initialize with pretrained | initialize randomly      | 410               | 41              | 100              | 551            | 72.5     |
> > | ResNet-50  | N              | initialize randomly        | initialize with searched | 0                 | 41              | 100              | 141            | 73.1     |
> > | ResNet-50  | Y              | initialize with pretrained | initialize with searched | 410               | 41              | 100              | 551            | 73.8     |
> > | ResNet-152 | N              | initialize randomly        | initialize with searched | 0                 | 116             | 100              | 216            | **74.6** |
> >
> >
> > There are three stages involved in the table.
> >
> > 1. Pretrain stage (optional): train the supernet with random initialization. For example, we pre-train ResNet50/ResNet152 with 100 epochs in the table.
> > 2. Search stage: search(prune) from the supernet to a smaller model. The searched model, including a structure and a set of weights, is obtained in this stage. For example, we prune ResNet50/ResNet152 to models with 1G flops using 10 epoch in the table.
> >
> >    There are two ways to initialize the supernet for searching.
> >    1. initialize with pretrained weights from the pretrain stage.
> >    2. initialize randomly. When we initialize randomly, we do not need pretrain stage. Note, that we jointly optimize the model parameters and structure parameters in the search stage. So even if we initialize the supernet randomly, the search stage can still output a set of weights, which is better than random initialization.
> > 3. Retrain stage: train the searched model, from the search stage. For example, we retrain the searched models with 100 epochs in the table.
> >
> >    There are two ways to initialize the searched model.
> >    1. initialize with the weights of the searched model from the search stage.
> >    2. initialize randomly. It only keeps the structure of the searched model, and initializes the weights randomly.
> >
> > From the row 4,5 of the table, we can see searching from a randomly initialized ResNet152 is better than searching from a pretrained ResNet50, and the former costs less resource.
> >
> > We hope this table can help you understand our method better. If you have any other questions, please indicate which stage or which initialization scheme make you confused. It will be helpful for us to answer your questions.
> >
> > > Q8  Unfortunately, I can't conclude that the method would have ...
> >
> > We are conducting experiments to compare our method with ZiCo in the same training setting. But it will take some time to finish the experiments. We will comment on this question after we finish the experiments, again.
> >
> >
> > Thanks for your feedback, again. It makes our paper better. If you have further questions, please let us know.
> > We expect to describe our method more clearly to address your question, especially w2, q6, and q7, as they are important to our method.
> >
> > Looking forward to your reply.

---

> > ### Author Response · Authors · 2023-11-23
> > **Additional response to "Rebuttal Response"**
> >
> > > Q8: Unfortunately, I can't conclude that the method would have ...
> >
> >
> > We apologize for not finishing this experiment completely before the rebuttal deadline.
> > Here, we provide a preliminary result, in which we train the models with the same training script with 1/16 ImageNet-1K dataset.
> > | Method | Top-1 |
> > | ------ | ----- |
> > | ZiCo   | 64.4  |
> > | DMS    | 68.6  |
> >
> >
> > Besides, the experiment on full ImageNet has run 32 epochs in 150 epochs. The results are shown in the table below.
> >
> > | Method | Top-1 |
> > | ------ | ----- |
> > | ZiCo   | 48.1  |
> > | DMS    | 56.3  |
> >
> >
> > It can be seen that our method outperforms ZiCo by a large margin. We will add the result of the experiment to our paper after the experiment is finished.
> >
> > Although we cannot provide the complete result of comparison with ZiCo in the same training setting, the performance gain of our method is still convincing, as it provides stable improvement on different models and datasets.
> > We hope you can consider more about the novelty and high search efficiency of our method, and we would appreciate it if you could consider raising the score of our paper. Thank you very much.

---

### Official Review · Reviewer_AHK9 · 2023-11-02

**Soundness:** 3 good
**Presentation:** 4 excellent
**Contribution:** 3 good
**Rating:** 6
**Confidence:** 4

**Summary:**

This paper introduces Differential Model Scaling (DMS) to increase the efficiency of width and depth search in networks. The differential top-k introduces by the method to model structural hyperparameters in direct and differentiable manner lays the foundation of this approach. The method is evaluated fairly exhaustively on different image classification architectures like EfficientNet-B0, DeiT on ImageNet. Furthermore the method is also evaluated on myriad tasks like object detection on COCO and language modelling (with Llama-7B model). The proposed method achieves significant improvements over different NAS baselines and some handcrafted architectures.

**Strengths:**

- The approach presented is very novel and well motivated.
- Experimental evaluation (across different scales, applications, model variants) is exhaustive. The paper also ablates the initialisation scheme of the architecture thoroughly. The search time comparison between different methods is also provided, thus showing the compute savings of the method.
- The presentation is clear and the paper is well written.
- The contribution of the paper is very significant especially since it scales NAS methods to realistic search spaces.

**Weaknesses:**

- Search time comparison in some cases seems unfair/confusing (refer to questions)
- Since the problem is cast as a NAS problem the search spaces used are not the ones very traditional to NAS (refer to questions)
- The search needs to be repeated for every resource constraint and obtaining a Pareto-Front of objectives might be very expensive (unlike methods like OFA[1]  which directly approximate the whole Pareto-front)

**Questions:**

- Search time comparison -> Since the observation from section 5 show that initialization from pre-trained models is very useful for differential scaling, did the authors include this in the search time computation. If a method relies on pre-trained models, then ideally the pre-training cost is a part of the total cost incurred. Could the authors clarify the intialization scheme used in each of the tables ie. table 1,2,3.
- In the appendix the authors compare with one-shot methods like OFA [1] . The comparison in my opinion is unfair since the search is performed on different search spaces. Could the authors evaluated the method on the exact same search space as OFA? This would help differentiable the gains of the search-space v/s the method itself? Similarly  could a comparison be made with the AutoFormer [2] by evaluating the method on its exact search space [2]?

I am willing to increase my score if my concerns are addressed as I believe this is a very interesting and impactful work.

[1] Cai, H., Gan, C., Wang, T., Zhang, Z. and Han, S., 2019. Once-for-all: Train one network and specialize it for efficient deployment. arXiv preprint arXiv:1908.09791.

[2]Chen, M., Peng, H., Fu, J. and Ling, H., 2021. Autoformer: Searching transformers for visual recognition. In Proceedings of the IEEE/CVF international conference on computer vision (pp. 12270-12280).

---

> ### Author Response · Authors · 2023-11-16
> **Response to  Reviewer AHK9**
>
> Dear reviewer:
>
> Thanks for your nice comments and suggestions. We have revised the paper according to your suggestions. The following are our responses to your comments.
>
> > W1: "Search time comparison in some cases seems unfair/confusing (refer to questions)"
> > Q1: "Search time comparison -> Since the observation from section 5 show that initialization from pre-trained models is very useful for differential scaling, did the authors include this in the search time computation. If a method relies on pre-trained models, then ideally the pre-training cost is a part of the total cost incurred. Could the authors clarify the intialization scheme used in each of the tables ie. table 1,2,3."
>
> We are sorry for the confusion.
> To clarify, we did not load pretrained models by default. We only load pretrained models in Table 3 (LLM experiments), Table 4 (ablation study), and Table 8 (comparison with other pruning methods). Therefore, we did not include the pretraining cost in Table 1 for our method as we did not use pretraining, and we think the comparison is fair for other methods.
> We chose this setting in experiments because, based on our ablation study, we found searching from a large, randomly initialized supernet reaches higher accuracy but lower cost than searching from a small pretrained supernet. Details are in the table below.
>
> | Supernet | $Iinit_{search}$ | $cost_{pretrain}$ | $cost_{search}$ | $cost_{total}$ | Top-1 |
> |----------|------------------|-------------------|-----------------|----------------|-------|
> | ResNet-50 | Random | 0 | 41 | 41 | 73.1 |
> | ResNet-50 | Pretrain | 410 | 41 | 451 | 73.8 |
> | ResNet-152 | Random | 0 | 116 | 116 | **74.6** |
>
> We have updated this table to section 5 to make our conclusion more clear.
>
> > W2: "Since the problem is cast as a NAS problem the search spaces used are not the ones very traditional to NAS (refer to questions)"
> > Q2: "In the appendix the authors compare with one-shot methods like OFA [1] . The comparison in my opinion is unfair since the search is performed on different search spaces. Could the authors evaluated the method on the exact same search space as OFA? This would help differentiable the gains of the search-space v/s the method itself? Similarly could a comparison be made with the AutoFormer [2] by evaluating the method on its exact search space [2]?"
>
> There are two main differences between the search space of our method and that of OFA and Autoformer.
>
> 1. What is more fine-grained in OFA's search space: We (and Autoformer) only search for the depth and width of networks, while OFA also searches for the resolution and kernel size. This is because we want to build a general method that can be applied to NLP and CV tasks. The resolution and kernel size are not necessary for NLP tasks.
> 2. What is more fine-grained in our search space: We can search width and depth with step 1 due to our high search efficiency. However, OFA and Autoformer have to use large steps and min-max limitations to compress their search space. This is an advantage of our method, as our search space is easier to define and much larger.
>
> Our current submission uses the default setting directly, causing different search spaces. To make the comparison more fair, we select Autoformer-T as our baseline and use the exact same supernet and search space from Autoformer-T. The performance and estimated search cost are shown below.
>
> | Method | Search Space | MACs | Top-1 | $Cost_{search}$ |
> |--------|--------------|------|-------|-----------------|
> | Autoformer-T | AutoFormer | 1.3 G | 74.7 | > 25 GPU days |
> | DMS | Autorormer | 1.3 G | **75.2** | 2 GPU days |
>
> Using Autoformer's search space, our method outperforms Autoformer-T by 0.5%  with less than 10% of the search cost. It proves the gain of our search method is from itself.
>
> We have also added this ablation study to Appendix A.5.2 in our revised paper.
>
> > W3: The search needs to be repeated for every resource constraint and obtaining a Pareto-Front of objectives might be very expensive (unlike methods like OFA[1] which directly approximate the whole Pareto-front).
>
> Yes, we agree with your opinion. Our method is not suitable for searching for the Pareto-Front.
>
> However, we want to emphasize our target is to build a general and flexible NAS method. There are several situations where our method is more suitable than OFA.
>
> 1. Searching for a small number of models, which is the usual case in the real world.
> 2. Searching extremely large models. For example, we prepare to train an LLM with 100B parameters using all our GPUs for 30 days. It's impossible to train a supernet (by OneShot NAS, like OFA), which needs much over 30 days, such as 60 or even 300 days. But using much less than 30 days, such as 1 day or 3 days , to search for it with our method is acceptable.
>
> Last, thank you for your helpful reviews! Looking forward to hearing back from you!

---

> ### Author Response · Authors · 2023-11-21
> **Sincerely expecting feedback from reviewer AHK9.**
>
> Dear reviewer:
>
> Thanks for your constructive comments. We have posted our responses to your comments. We expect your feedback about whether our responses address your concerns, or if you have any further questions. We are glad to answer them and improve our paper.
>
> Best,
>
> Authors

---

### Author Response · Authors · 2023-11-16
**Response to all Reviewers**

Dear Reviewers,

Thank you for your valuable comments. We have responded to each of your comments and revised our paper accordingly. Here, we summarize the major changes in the revised version.
1. We add four new ablation studies to Appendix 5, including
   1. About search space (compare our method with Autoformer using the exact same search space)
   2. About element importance metric (compare a basic metric, index metric, and Taylor importance with/without a moving average)
   3. About our hyperparameters that need to be tuned for different models and tasks.
2. Some modifications are made to make our paper more clear.
   1. Update tables in section 5 to make our conclusion clearer.
   2. Add initialization scheme description to all tables.
   3. Update Appendix A.1.1 to make how our method works with different layers more clearly.
   4. Fix typos and make other modifications to make our paper more clear.

Thanks for your constructive comments, which help us improve our paper. If you have any further questions, please feel free to comment. We will reply as soon as possible.
Besides, our paper proposes a general and flexible NAS method, which makes the NAS more practical for real-world applications. We believe our work is valuable and deserves to be published. Therefore, we hope you can raise the score of our paper if our responses address your concerns. Thank you very much.

---

### Meta-Review · Area_Chair_KwDk · 2023-12-05

**Metareview:**

The paper proposes a new differentiable architecture search method, namely Differential Model Scaling (DMS) to automatically scale width and depth in networks. DMS directly searches the  width and depth of an network via an differential top-k Differential Model Scaling which is based on  Taylor importance. Then  the authors test DMS on several representative  tasks, e.g., image classification, object detection, and language modelling.

The main strength of this work is that its proposed Differential Model Scaling (DMS) is totally differentiable and can be easily optimized. However, almost all reviewers emphasize the insufficient experiments. For example, three reviewers who give positive scores still think that 1) not very fair  experiments, e.g., time comparison, 2) missing comparison of memory cost and latency; 3) missing comparison of some important baselines, e.g., element importance methods; and 4) ablation study does not well show the effect of the differential topk. Another two reviewers who have negative scores claimed the improvement is incremental. For this point, I partly agree. Moreover, the reported classification accuracy of ResNet50 on ImageNet is often around 75% under conventional supervised setting, and is higher than the accuracy of the searched network in this work.  Consider all these experimental factors together, we cannot accept this work in this time. The authors can accordingly solve these issue for improvement.

**Justification For Why Not Higher Score:**

Almost all reviewers emphasize the insufficient experiments. For example, three reviewers who give positive scores still think that 1) not very fair  experiments, e.g., time comparison, 2) missing comparison of memory cost and latency; 3) missing comparison of some important baselines, e.g., element importance methods; and 4) ablation study does not well show the effect of the differential topk. Another two reviewers who have negative scores claimed the improvement is incremental. For this point, I partly agree. Moreover, the reported classification accuracy of ResNet50 on ImageNet is often around 75% under conventional supervised setting, and is higher than the accuracy of the searched network in this work.

Consider all these experimental factors together, we cannot accept this work in this time.

**Justification For Why Not Lower Score:**

N/A

---

### Decision · Program_Chairs · 2024-01-16

Reject